# TIPS: TURN-LEVEL INFORMATION-POTENTIAL REWARD SHAPING FOR SEARCH-AUGMENTED LLMS

**Yutao Xie**[1]  **Nathaniel Thomas**[1]  **Nicklas Hansen**[1]  **Yang Fu**[1]

**Li Erran Li**[2]  **Xiaolong Wang**[1]

[1]UC San Diego  [2]AWS AI

## ABSTRACT

Search-augmented large language models (LLMs) trained with reinforcement learning (RL) achieve strong results on open-domain question answering (QA), but training remains brittle: rewards are sparse, credit assignment across reasoning and tool calls is difficult, and optimization often collapses on long-horizon tasks. We introduce **Turn-Level Information Potential Reward Shaping (TIPS)**, a simple RL framework that assigns dense rewards to each reasoning–tool-call segment based on how much it increases a teacher model's log-likelihood of the correct answer. This potential is computed by a frozen or periodically refreshed copy of the policy, so TIPS only requires checkpoints of the model being trained—no separate reward model, verifier, or human process labels—making it practical for scaling to frontier models. We show that this turn-level information reward is a form of potential-based shaping, preserving the task's optimal policy while providing fine-grained guidance beyond outcome-only supervision. On a search-augmented QA setting spanning seven in-domain and out-of-domain benchmarks, TIPS consistently outperforms PPO/GRPO baselines and substantially improves training stability; for example, on Qwen-2.5-7B Instruct it improves average Exact Match by 11.8% and F1 by 13.6% over PPO. These results suggest that information-potential shaping is a viable general mechanism for stabilizing long-horizon RL on large, tool-using LLMs. The code base for TIPS is available at https://github.com/ucsd-wang-lab-lm/tips.

## 1 INTRODUCTION

Large language models (LLMs) have recently shown substantial improvements in reasoning when fine-tuned with reinforcement learning (RL) under *verifiable* rewards (Jaech et al., 2024; Guo et al., 2025; Lambert et al., 2024). In tool-using settings, this paradigm is especially appealing: we can define correctness at the end of an interaction (e.g., the final answer matches a reference), while allowing the model to freely interleave reasoning with retrieval, code execution, or database queries. Yet this same property exposes a central bottleneck: the learning signal is often *outcome-only*—a single terminal reward after a long sequence of reasoning and tool calls.

**Outcome-only RL is brittle for training tool-use LLMs.** Outcome-only rewards induce a severe *credit assignment* problem: the agent must infer which intermediate decisions caused success or failure, despite receiving feedback only at the end (Ng et al., 1999; Devlin & Kudenko, 2012; Wiewiora, 2011). In multi-turn tool use, credit assignment is harder than in pure chain-of-thought reasoning for two structural reasons. First, each tool call is a discrete intervention that changes the agent's information state: a retrieval result, an execution trace, or a table slice can fundamentally alter what is knowable in later turns. Thus, the policy does not only choose *how* to reason, but also *what information* to acquire and when. Second, many distinct intermediate trajectories are consistent with the same terminal outcome (underspecification): a correct final answer may be reachable through both helpful and redundant tool calls, while an incorrect final answer might arise from a single early mistake that later steps cannot repair. As a result, the terminal reward provides little guidance on which earlier turns were informative versus distracting, leading to high-variance updates and fragile learning dynamics (e.g., policy collapse or drift) when optimizing long-horizon behaviors with

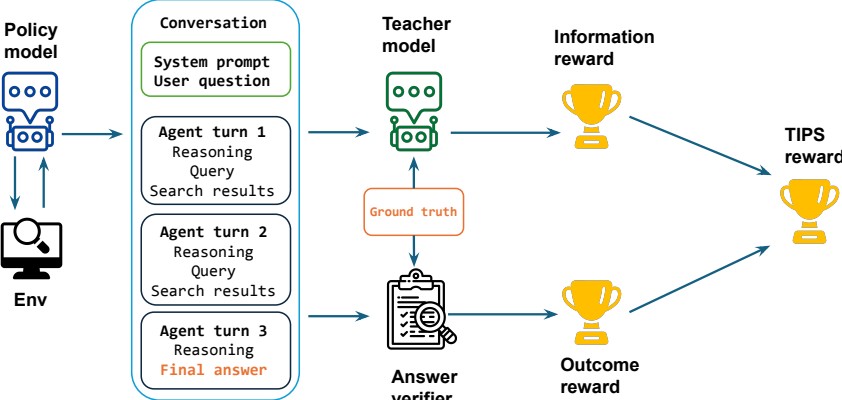

*Figure 1.* **Overview of our training framework.** The policy model interacts with the environment by conducting multi-turn conversations: each turn consists of reasoning, issuing a query, and receiving search results, until a final answer is produced. Two dense reward signals are then derived: (*i*) an **outcome reward**, obtained by verifying whether the final answer matches the ground truth; and (*ii*) an **information reward**, provided by a teacher model that measures the information gain each turn contributes toward the ground truth. Both rewards are combined and optimized with PPO.

sparse feedback (Arjona-Medina et al., 2019). This issue is particularly acute in open-domain question answering (QA), where a binary correct/incorrect signal does not indicate which intermediate retrievals or tool invocations improved evidence quality (Lightman et al., 2023; Gurung et al., 2025; Zeng et al., 2025b), thereby under-incentivizing reliable multi-turn reasoning with tools.

A natural response is to densify supervision. Process supervision and process reward models (PRMs) provide token- or step-level rewards that better shape intermediate reasoning (Lightman et al., 2023; Wang et al., 2024). While effective, these methods typically require high-quality intermediate labels and/or substantial offline training of reward models, and their supervision granularity is not always aligned with tool-use interactions, where the semantically meaningful unit is a *turn* consisting of reasoning, a tool action, and the resulting observation. Another direction assigns turn-level rewards from environment feedback. For example, MT-GRPO uses turn-level signals derived from tool outcomes, but it was designed for a single tool call; when extended to multiple calls, the signal becomes less discriminative and can inherit the instability of sparse-reward baselines (Table 1). Overall, prior work reveals a gap: we want turn-level feedback that is (i) *discriminative* enough to distinguish helpful from unhelpful tool interactions, (ii) *lightweight* (no token-level labeling or training an additional reward model), and (iii) *compatible* with standard RL fine-tuning.

We propose **TIPS, a turn-level credit assignment method based on *information gain***. We represent an interaction as a sequence of turns, each following the Thought–Action–Observation pattern common in tool-using agents (Yao et al., 2022; Nakano et al., 2021; Schick et al., 2023): a segment of reasoning tokens, a tool invocation, and the tool output. ***Intuitively, a good turn is one that makes the correct answer* more predictable *given the accumulated context, while a distracting turn does not.*** Concretely, we quantify the contribution of a turn by measuring the *incremental* increase in the agent's log-likelihood of the gold final answer when that turn is appended to the trajectory. ***Turns that raise the likelihood receive positive credit; turns that do not help (or mislead) receive small or negative credit.*** This converts an outcome-only signal into dense, turn-level feedback that directly targets the core ambiguity in tool-use credit assignment: which information-acquisition steps actually moved the agent closer to a correct answer.

We formalize the interaction as a *segment-level* Markov decision process (MDP), where each action corresponds to choosing a tool invocation (and its associated reasoning) and the environment returns the tool observation. Within this view, our information credit takes the form of a *potential difference* over segment states, yielding a potential-based reward shaping (PBRS) construction (Ng et al., 1999; Devlin & Kudenko, 2012; Wiewiora, 2011). Under standard conditions, PBRS preserves the set of optimal policies while improving learning signals, offering a clean interpretation of TIPS as a policy-invariant way to reduce variance and stabilize long-horizon optimization. We integrate the shaped reward into PPO (Schulman et al., 2017) for direct optimization in standard RL workflows, and study how some internal design choices affect the training.

We evaluate TIPS across eight in-domain and out-of-domain QA benchmarks with multi-turn tool use. Compared to strong PPO/GRPO baselines that rely on verifiable outcome rewards (Zeng et al., 2025a; Su et al., 2025), TIPS yields consistent gains and more stable training dynamics. On Qwen-2.5-7B Instruct, TIPS improves average exact match (EM; Appendix E.1) and F1 (Appendix E.2) by over 10% relative to PPO, with larger margins over GRPO, while exhibiting reduced training variance and fewer collapse/drift failures. These results highlight that turn-level information shaping is an effective, lightweight alternative to token-level process supervision for making tool-using agents trainable under sparse terminal rewards.

**Our contributions can be summarized as follows:** *First*, we introduce **TIPS**, a reinforcement learning framework for multi-turn LLM agents that models trajectories as segment-level MDPs and assigns information-gain rewards to each turn. *Second*, we integrate this turn-level reward shaping into PPO using potential-based reward shaping, which preserves policy invariance and stabilizes long-horizon optimization. *Third*, we validate TIPS on search-augmented open-domain QA across 8 benchmarks and observe consistent gains over PPO and GRPO, with the strongest improvements on multi-hop and out-of-domain benchmarks. *Fourth*, we analyze training dynamics and advantage distributions, and demonstrate that TIPS delivers more stable learning and allocates credit to effective reasoning and tool use while discouraging degenerate behavior.

## 2 PRELIMINARIES

**Problem formulation.** We consider an LLM-based search agent for question answering. The agent may invoke a natural language search tool—embedding-based or keyword retrieval—that ingests a query string and returns the top-k passages. Given a dataset $\mathcal{D} = \{(x_i, \mathcal{A}_i)\}$ where each question $x_i$ has a small gold answer set $\mathcal{A}_i$, the agent must produce a response through iterative reasoning and retrieval. Critically, the response must embed its final answer within `<answer>` tags, and evaluation extracts this tagged answer to compute exact match (EM) or F1 against $\mathcal{A}_i$; this is not an open-ended generation task.

**Token-level MDP.** We model the interaction as a finite-horizon Markov Decision Process (MDP) at the token level. At step $t$, the state $s_t$ consists of the prefix of all generated text so far including question, reasoning history and retrieved evidence. The action $a_t$ is the next token sampled from the policy $\pi_\theta(a_t \mid s_t)$. To transition to the next state, we either concatenate $s_{t+1} = s_t \oplus a_t$, or if $a_t$ triggers a retrieval, we append an observation $O_k$, defining a boundary index $b_k$. The episode terminates when an EOS token is emitted, yielding a final reward $R_{\text{final}}$ based on answer correctness. All other tokens are assigned reward 0.

**On-policy RL with a response mask.** We focus on PPO and GRPO, the most common on-policy RL baselines for LLMs. PPO trains $\pi_\theta$ against its snapshot $\pi_{\text{old}}$ with clipped importance ratios

$$\rho_t(\theta) = \frac{\pi_\theta(a_t \mid s_t)}{\pi_{\text{old}}(a_t \mid s_t)}, \qquad \mathcal{L}_{\text{clip}}(\theta) = \mathbb{E}_t\big[m_t \min(\rho_t(\theta)A_t, \ \text{clip}(\rho_t(\theta), 1 - \varepsilon, 1 + \varepsilon)A_t)\big], \quad (1)$$

where $A_t = G_t - V_\phi(s_t)$, $G_t = \sum_{k=t}^{T-1} \gamma^{k-t} r_k$, $r_t$ is the per-token reward, and $V_\phi$ is a learned value baseline. GRPO removes the critic by normalizing sequence-level rewards: given $g$ rollouts with terminal rewards $\{R^{(i)}\}$, compute mean $\mu$ and stdev $\sigma$, then set $A^{(i)} = (R^{(i)} - \mu)/\sigma$. A binary mask $m_t \in \{0, 1\}$ excludes non-trainable tokens (e.g., prompts or tool outputs).

## 3 METHOD

**Credit assignment is the bottleneck in multi-turn tool use.** In tool-augmented QA, the model must decide not only *how* to generate text, but also *when* and *what* information to acquire via tools. The essential credit assignment question is therefore *turn-level*: among the turns that change the agent's information state (via retrieved evidence or tool outputs), which turns actually made a correct answer more attainable, and which were redundant or misleading. Sparse terminal rewards provide little supervision for this: a model receives no feedback on which intermediate tool calls are beneficial until episode completion, and many distinct turn sequences can lead to the same final outcome. A correct answer may be reached despite unnecessary tool calls, while an incorrect answer may be caused by a single early mistake that later turns cannot repair. This many-to-one mapping

from multiple actions and CoTs to a single outcome reward yields high-variance learning signals and brittle optimization in long-horizon training (Fig. 3).

TIPS addresses this by using the policy itself as a teacher: a frozen snapshot of the current policy scores each turn by measuring the increase in its own log-probability of generating a correct answer. Intuitively, a turn that retrieves a highly relevant passage will make the teacher much more confident in some answer in $\mathcal{A}$, whereas a redundant or off-topic query leaves its belief nearly unchanged or even shifts probability mass toward incorrect answers. The change in log-probability thus provides a single scalar measure of how much turn $k$ helped move the dialogue toward a correct answer. Because the teacher is a lagged copy of the policy, their predictive distributions are kept close, so what elevates the teacher's confidence tends to benefit the policy as well. We periodically refresh the teacher to prevent its beliefs from becoming stale and misaligned with the latest policy. Besides, the mechanism requires no external judge and only adds a modest computational overhead.

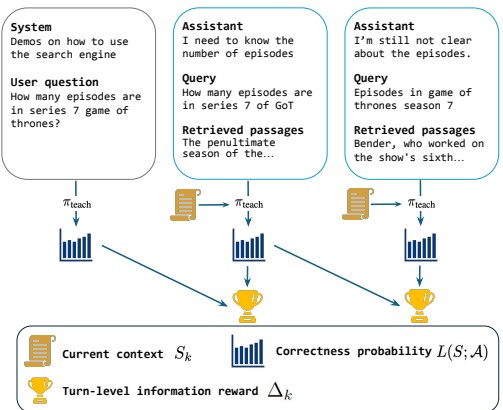

*Figure 2.* **Turn-level information reward pipeline.** At each turn, retrieved evidence updates the answer likelihood, yielding a turn-level reward $\Delta_k$. These rewards are then injected at turn boundaries.

## 3.1 TURN-LEVEL INFORMATION REWARDS

In the multi-turn QA task, each turn comprises a reasoning block, a tool call, and the evidence returned. To formalize the intuition above, let $\mathcal{A} = \{A^{(1)}, \ldots, A^{(M)}\}$ be the set of valid answers for an episode. For any context $S$, we define the *answer potential*:

$$\Phi(S) := L(S; \mathcal{A}) = \log \sum_{m=1}^{M} p_{\text{teach}}(A^{(m)} \mid S),$$

the teacher's log-probability of generating *any* correct answer. The turn-level reward for turn $k$ is the change in this potential:

$$\Delta_k = \alpha \big[ \Phi(S_k) - \Phi(S_{k-1}) \big],$$

where $S_k$ is the context up to and including turn $k$ and $\alpha > 0$ scales the reward. Intuitively, $\Phi(S)$ summarizes, under the teacher, how likely the current dialogue state is to yield any correct answer in $\mathcal{A}$, and $\Delta_k$ measures how much turn $k$ moves this likelihood up or down. This quantity is positive when turn $k$ shifts probability mass toward valid answers, and negative otherwise.

**Information-theoretic interpretation.** $\Delta_k$ can be interpreted as a scaled pointwise mutual-information quantity between the evidence at turn $k$ and answer correctness under the teacher's distribution: it measures how much information the new observation contributes about the event that the answer lies in $\mathcal{A}$. This view is especially natural for QA, where each tool call is intended to supply incremental evidence toward a correct answer.

## 3.2 SEGMENT-LEVEL PBRS AND POLICY INVARIANCE

So far, $\Delta_k$ has been defined as a dense, turn-level signal that measures how much turn $k$ helps the teacher believe in a correct answer. We now show that this signal can be viewed as a standard potential-based reward shaping term, and therefore does not change the optimal policy under the original outcome reward. We treat each turn as a segment of tokens between turn boundaries, and view a whole turn as a single action in a segment-level MDP (where the action subsumes reasoning, tool calls, and observations). With the answer potential $\Phi(S)$ from Section 3.1 as the potential function, the shaping reward at turn $k$ is exactly

$$\Delta_k = \alpha \big[ \Phi(S_k) - \Phi(S_{k-1}) \big],$$

which matches the standard form of potential-based reward shaping (Ng et al., 1999).

**Policy invariance guarantee.** Under episodic returns ($\gamma = 1$) and shaping confined to segment boundaries, the shaped Monte Carlo return for any token $t$ in turn $k$ satisfies:

$$G_t^{(R+I)} = G_t^{(R)} + \sum_{j=k}^{K} \Delta_j = G_t^{(R)} - \alpha\,\Phi(S_{k-1}), \tag{2}$$

where we assign $\Phi(S_K) = 0$. Thus $G_t^{(R+I)}$ differs from $G_t^{(R)}$ only by a constant that depends on the segment boundary state $S_{k-1}$, and is independent of the within-turn action sequence $\tau_k$. Actions that are relatively better under the outcome reward therefore remain relatively better after shaping. As a result, policy improvement under Monte Carlo estimation—and hence under GAE or other policy-gradient estimators that use these returns up to a baseline—is unaffected.

**Advantage Baseline Calibration via TIPS.** Since $\sum_{j=k}^{K} \Delta_j = -\alpha\,\Phi(S_{k-1})$, every token inside turn $k$ receives the same offset determined only by the pre-turn state $S_{k-1}$. Hence, TIPS *densifies* learning signals and reduces ambiguity about which turns were helpful, while preserving the optimal policy under the terminal reward. A full proof is provided in Appendix F.

**Teacher re-syncing.** With a fixed teacher, our shaping is potential-based, so for any state the shaped Monte-Carlo return differs from the outcome-only return only by a state-dependent constant. When we refresh the teacher, we simply change the potential used for shaping in subsequent rollouts; in the PBRS view this is equivalent to changing a (state-dependent) baseline and leaves the true advantage function invariant, though with an approximate critic, we do observe small shifts in raw returns before it re-stabilizes in practice.

## 4 EXPERIMENTS

In our experiments, we seek to answer the following questions:

- **(Q1) Performance & stability.** Can TIPS train multi-turn QA agents that outperform outcome-only PPO/GRPO across in-domain and OOD tasks *and* remain stable (lower collapse rate, smoother/higher plateaus, reduced across-seed variance)?
- **(Q2) Learning signal design.** Does turn-level information reward improve credit assignment compared with sparse final-answer rewards and rule/rubric baselines? How do token-level advantage distributions differ under TIPS vs. PPO?
- **(Q3) Analysis & ablations.** How do shaping scale $\alpha$ and teacher freshness (frozen vs. periodic refresh) affect stability and final EM/F1? What are the comparative effects of dense-reward choices (rule-based, rubric/LLM-judge, information gain)?

**Experimental Setup.** Based on VeRL's multi-turn QA tasks setup, we conduct experiments with two model sizes: **Qwen-2.5-3B Instruct** and **Qwen-2.5-7B Instruct** (Jin et al., 2025; Qwen Team, 2025; Sheng et al., 2024). For retrieval, we use the E5 model (Wang et al., 2022) over the 2018 Wikipedia dump (Wikimedia Foundation, 2025), retrieving 3 passages per turn, and train on the merged NQ and HotpotQA training splits. We evaluate both in-domain and out-of-domain performance on seven QA benchmarks: NQ, TriviaQA, PopQA, 2WikiMultiHopQA, MuSiQue, HotpotQA, and Bamboogle (Kwiatkowski et al., 2019; Joshi et al., 2017; Mallen et al., 2023; Ho et al., 2020; Trivedi et al., 2022; Yang et al., 2018; Press et al., 2023). For all evaluations, we report Exact Match (EM) and F1 scores (Jin et al., 2025; Zhao et al., 2025).

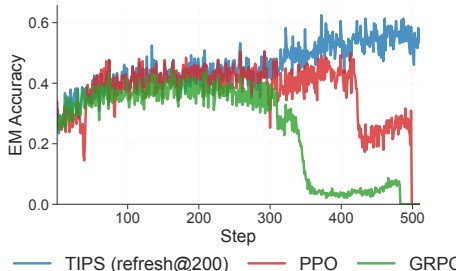

*Figure 3.* **EM accuracy on the training set.** TIPS converges to a high accuracy; PPO drifts late; GRPO collapses.

Unless specified, we use the following hyperparameters across all methods: PPO with GAE ($\lambda=1$, $\gamma=1$), KL penalty weight $\beta=0.001$, clip ratio $\varepsilon=0.2$, batch size = 256, context length = 4096, and maximum retrieval turns = 4. Training runs for 500 steps, or until a performance collapse to 0, on

*Table 1.* **Exact Match (EM) for 7B and 3B models.** In-domain: {NQ, HotpotQA}. Out-of-domain: {TriviaQA, PopQA, 2Wiki, MuSiQue, Bamboogle}. The last column is the average over all 7 tasks.

| Model | In-domain | | Out-of-domain | | | | | Avg |
|---|---|---|---|---|---|---|---|---|
| | NQ | HotpotQA | TriviaQA | PopQA | 2Wiki | MuSiQue | Bamboogle | |
| **Qwen-2.5-7B Instruct** | | | | | | | | |
| PPO | 41.95 | 34.46 | 63.71 | 43.55 | 32.94 | 8.94 | 35.40 | 37.28 |
| GRPO | 37.15 | 26.54 | 56.39 | 37.11 | 19.41 | 7.20 | 16.00 | 28.54 |
| MT-GRPO* | 37.17 | 29.28 | 58.29 | 38.37 | 22.62 | 7.99 | 19.20 | 30.42 |
| MT-PPO | 42.37 | 26.45 | 55.12 | 41.59 | 22.81 | 6.91 | 11.20 | 29.49 |
| TIPS | **43.38** | **42.95** | **64.31** | **44.52** | **42.96** | **17.05** | **36.80** | **41.71** |
| **Qwen-2.5-3B Instruct** | | | | | | | | |
| PPO | **43.80** | 27.12 | 58.28 | **42.81** | 23.10 | 6.37 | 9.60 | 30.15 |
| GRPO | 37.40 | 29.62 | 55.23 | 37.39 | 26.85 | **8.73** | **20.80** | 30.86 |
| MT-GRPO* | 36.18 | 24.71 | 51.70 | 35.64 | 22.63 | 5.17 | 10.40 | 26.35 |
| MT-PPO | 39.65 | 25.54 | 56.17 | 40.47 | 21.35 | 6.16 | 8.00 | 28.19 |
| TIPS | 43.46 | **31.40** | **58.80** | 42.76 | **29.25** | **8.73** | **20.80** | **33.60** |

*Table 2.* **F1 for 7B and 3B models.** In-domain: {NQ, HotpotQA}. Out-of-domain: {TriviaQA, PopQA, 2Wiki, MuSiQue, Bamboogle}. The last column is the average over all 7 tasks.

| Model | In-domain | | Out-of-domain | | | | | Avg |
|---|---|---|---|---|---|---|---|---|
| | NQ | HotpotQA | TriviaQA | PopQA | 2Wiki | MuSiQue | Bamboogle | |
| **Qwen-2.5-7B Instruct** | | | | | | | | |
| PPO | 50.53 | 44.65 | 70.88 | 47.62 | 38.70 | 17.79 | 45.33 | 45.07 |
| GRPO | 47.85 | 35.29 | 63.58 | 42.51 | 24.15 | 12.12 | 22.93 | 35.49 |
| MT-GRPO* | 48.10 | 38.36 | 66.05 | 44.11 | 27.94 | 14.29 | 28.25 | 38.16 |
| MT-PPO | 50.60 | 35.29 | 62.08 | 46.45 | 28.56 | 12.13 | 20.87 | 36.57 |
| TIPS | **53.22** | **54.66** | **72.17** | **49.26** | **50.64** | **26.58** | **52.16** | **51.24** |
| **Qwen-2.5-3B Instruct** | | | | | | | | |
| PPO | 51.71 | 36.47 | 65.75 | 46.81 | 28.73 | 12.65 | 19.12 | 37.32 |
| GRPO | 47.40 | 39.51 | 63.30 | 43.00 | 33.36 | 14.81 | 28.45 | 38.55 |
| MT-GRPO* | 45.77 | 33.10 | 59.51 | 40.40 | 28.13 | 10.68 | 15.90 | 33.64 |
| MT-PPO | 47.75 | 34.15 | 62.85 | 44.83 | 26.10 | 11.05 | 14.00 | 34.39 |
| TIPS | **51.77** | **41.47** | **66.43** | **47.43** | **35.10** | **15.85** | **29.82** | **41.12** |

$8\times$H200 GPUs with FSDP and gradient checkpointing. Full hyperparameters of both training and search engine server setup are described in Appendix E.3.

**Baselines.** We compare against four related reinforcement learning baselines. **PPO** (Proximal Policy Optimization): Our base method with outcome-only supervision. Policy and value learning rates are $1 \times 10^{-6}$ and $1 \times 10^{-5}$, respectively. **GRPO** (Group Relative Policy Optimization): We follow the public implementation with group size 5 and learning rate $5 \times 10^{-7}$. To ensure stability, we apply gradient clipping at $1 \times 10^{-4}$, which mitigates much of the collapse observed in reproducing Search-R1. **MT-GRPO*** and **MT-PPO**: Multi-turn extensions of GRPO and PPO for multi-hop QA. MT-GRPO* propagates stepwise normalization across multiple tool calls, while MT-PPO applies environment feedback at turn–level. Both variants are tested with two rule-based rewards: (*i*) tool-correctness only, and (*ii*) correctness plus answer appearance in retrieved passages. In our main tables, we report the stronger variant for each model size: tool-only for 7B (answer-based often collapsed) and answer-aware for 3B. Implementations are in Appendix E.5.

*Table 3*. Generalization of TIPS across model families and scales. EM/F1 are TIPS scores; percentages in parentheses indicate relative improvement over the outcome-only PPO baseline. FLOPs overhead is the relative per-step increase due to teacher scoring.

| Model | EM | F1 | FLOPs overhead (%) |
|---|---|---|---|
| Qwen2.5-3B-Instruct | 33.6 (+11.4%) | 41.1 (+10.2%) | 11.761 |
| Qwen3-4B-Instruct-2507 | 48.4 (+7.3%) | 57.1 (+6.1%) | 11.846 |
| Qwen2.5-7B-Instruct | 41.7 (+11.9%) | 51.2 (+13.7%) | 11.810 |
| Qwen2.5-14B-Instruct | 45.4 (+12.7%) | 53.1 (+10.6%) | 11.813 |
| Llama3.1-8B | 40.3 (+34.0%) | 49.0 (+29.3%) | 11.659 |

## 4.1 MAIN RESULTS

From Tables 1 and 2, we find that **TIPS consistently outperforms all baselines**, with clear advantages over PPO and GRPO across both in-domain and out-of-domain settings. The largest absolute improvements appear on multi-hop out-of-domain tasks such as 2Wiki, MuSiQue, and Bamboogle, where outcome-only methods struggle. At the 3B scale, improvements are more modest and in some cases PPO performs competitively, whereas at the 7B scale TIPS shows a pronounced advantage.

For the MT baselines, we tested two rule-based reward variants (tool-correctness vs. answer-aware) and observed instability: on 7B models the answer-aware variant often collapsed, while on 3B models the tool-only variant underperformed. As a result, we report the stronger variant for each case in the tables. Overall, MT-GRPO* improves over GRPO by enabling multi-tool credit assignment, but both MT baselines still lag significantly behind TIPS. Combined with the training curves in Fig. 3, these comparisons highlight that TIPS not only achieves higher accuracy but also more stable optimization, avoiding the collapse of GRPO and the stagnation of PPO.

**Training dynamics.** As illustrated in Fig. 3, TIPS climbs steadily to an EM plateau of approximately 0.55-0.60 with low variance. In contrast, GRPO suffers a performance collapse around steps 320–350 and fails to recover, while PPO stagnates after 400 steps.

**Generalization across models.** Using the same training setup, we apply TIPS to five models from two families and multiple scales: Qwen2.5-3B/7B/14B, Qwen3-4B, and Llama3.1-8B. As summarized in Table 3, TIPS consistently improves over the corresponding outcome-only PPO baseline on all models, with relative EM gains ranging from +7.3% to +34.0% and F1 gains from +6.1% to +29.3%. The largest relative improvements appear on Llama3.1-8B, which starts from a weaker search capability and benefits most from better credit assignment, while stronger baselines such as Qwen3-4B still see solid gains. At the same time, the compute overhead of TIPS remains essentially constant across architectures: using the FLOPs accounting in Appendix H, teacher scoring adds only $\approx 11.7\%$ per-step FLOPs for all five models in Table 3. Taken together, these results support our claim that TIPS is backbone-agnostic, providing consistent improvements across model families at a modest and stable relative compute cost.

## 4.2 ANALYSIS

**Task-wise validation curves.** In Fig. 4, we plot EM curves of TIPS and PPO for Qwen2.5-7B across all benchmarks. Benchmarks are grouped into General QA (NQ, TriviaQA, PopQA) and Multi-hop QA (HotpotQA, 2Wiki, MuSiQue, Bamboogle), with the top-right panel showing the average across all seven. Results are computed on a held-out validation set of 6,000 samples with the same benchmark distribution as full evaluation. Across tasks, TIPS rises smoothly and quickly stabilizes, whereas PPO exhibits drift—most severe on multi-hop QA with mid-training degradation and only partial recovery. On general QA tasks the drift is milder but PPO still converges below TIPS. These dynamics align with Tables 1–2, where the largest margins appear on multi-hop/OOD datasets. Overall, TIPS delivers higher final EM and more reliable optimization by preventing PPO's late-stage collapse.

**Study of Advantage Distributions.** To further investigate contributors to TIPS' stability, we collect all unmasked token-level advantages from final checkpoints. In Fig. 5, TIPS yields a clean bimodal distribution with concentrated positive mass, while PPO shows fat-tailed positives and dense mass near zero, indicating instability and drift into poor policy space, which provides a mechanistic ex-

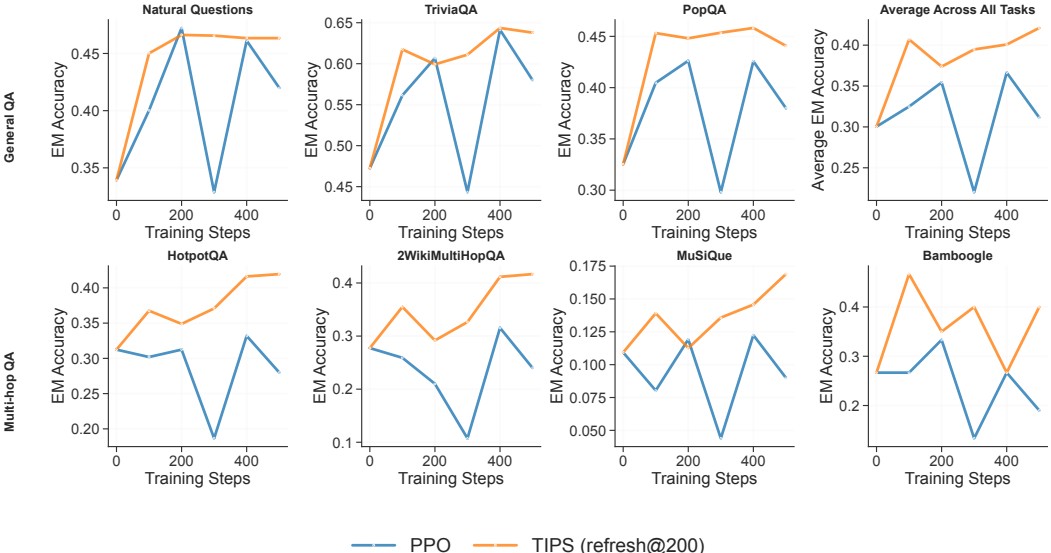

*Figure 4.* **Training dynamics of PPO vs. TIPS.** Blue curves denote PPO and orange curves denote TIPS with teacher refresh every 200 steps. Overall, TIPS climbs steadily to higher and more stable plateaus, while PPO often suffers mid-training drift or collapse, especially on multi-hop datasets.

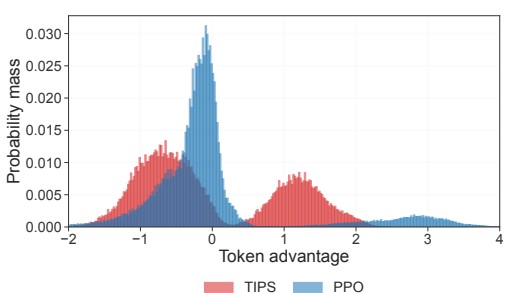

*Figure 5.* **Distribution of token-level advantages.** Aggregated advantages at final checkpoints. TIPS yields a clean bimodal distribution with concentrated positive mass, while PPO shows heavy tails and dense near-zero mass.

*Table 4.* **Ablations on dense credit sources.** Results are averaged over all tasks. MT-GRPO details are in Appendix E.5; the LLM-as-judge variant in Appendix E.7; MT-PPO uses the same reward design as MT-GRPO; and the history-max information gain variant in Appendix E.6.

| Method | EM | F1 |
|---|---|---|
| **GRPO family** | | |
| Outcome-only | 28.54 | 35.49 |
| Rule-based turn-level (MT-GRPO) | 30.42 | 38.16 |
| Rubric-based turn-level (LLM judge) | 28.23 | 35.54 |
| **PPO family** | | |
| Outcome-only | 37.28 | 45.07 |
| Rule-based turn-level (MT-PPO) | 29.49 | 36.57 |
| Turn-level information gain | 40.93 | 49.49 |
| History-max info gain | 35.20 | 43.09 |

planation for why **TIPS suppresses late-stage drift and collapse**, and how it stabilizes the training trajectory observed in Fig. 4.

**Computational Overhead.** We isolate the teacher-scoring cost by measuring per-step wall-clock time in a single TIPS run with and without scoring. This gives runtime overheads of $18\%$ for Qwen2.5-3B and $16\%$ for Qwen2.5-7B. In terms of FLOPs, TIPS adds only about $12\%$ relative to vanilla PPO for both model sizes, since we can reuse KV caches in the teacher forward passes on a given rollout. For comparison, GRPO requires roughly $3.5\times$ the PPO FLOPs for both model sizes. Raw wall-time differences between separate TIPS and PPO runs are heavily influenced by how long the model's responses are. A simple linear regression of per-step time on mean response length explains most of this gap, and after controlling for response length TIPS is within a few percent of PPO in wall-clock terms. Full overhead analysis is in Appendix H.

### 4.3 ABLATIONS

**Shaping scale $\alpha$.** The coefficient $\alpha$ controls the relative weight between the information reward and the terminal outcome reward. If $\alpha$ is too small, the shaping term becomes negligible and TIPS behaves like outcome-only PPO; if it is too large, the shaping term can compete with the terminal reward and increase gradient variance. In practice, we pick $\alpha$ so that the average per-turn informa-

*Table 5.* EM gains over PPO for different target information-reward ranges under dynamic $\alpha$. The medium band yields stable and consistently positive gains across backbones; very small targets effectively turn shaping off, while very large targets let shaping compete with the terminal reward and can hurt performance.

| Base model | Small (0.001–0.05) | Medium (0.05–0.3) | Large (0.3–1.0) |
|---|---|---|---|
| Qwen3-4B | +1.4% (stable) | +7.3% **(stable)** | −3.4% (stable) |
| Qwen2.5-7B | 0% (crashed) | +11.9% **(stable)** | +3.1% (stable) |
| Qwen2.5-3B | 0% (stable) | +11.4% **(stable)** | 0% (stable) |

*Table 6.* Ablation on teacher selection. Rows fix the policy backbone and vary the teacher.

| Policy | Frozen policy | Qwen3-4B-TIPS | Llama3.1-8B |
|---|---|---|---|
| Qwen2.5-7B | **41.7** (+11.9%) | 30.0 (-19.5%) | 29.0 (-22.2%) |
| Qwen3-4B | **48.4** (+7.3%) | 45.88 (+1.7%) | 43.0 (-4.7%) |

tion reward is clearly smaller than the terminal reward: we run a short pilot, estimate the typical magnitude of $|\Delta_k|$, and choose a fixed $\alpha$ such that $\mathbb{E}[|\alpha\Delta_k|] \approx 0.2$.

Across all backbones this places $\alpha$ in a medium band $\alpha \in [0.05, 0.3]$, within which TIPS is stable and consistently improves over outcome-only PPO. We also tried a dynamic-$\alpha$ scheme that keeps the mean information reward in a target band; as detailed in Appendix D, the medium band ($[0.05, 0.3]$) again yields stable and consistently positive gains, while very small or very large targets hurt performance.

**Different dense reward choices.** Table 4 compares different dense reward signals. Outcome-only rewards perform better under PPO than GRPO, reflecting the advantage of value-based credit assignment. Within GRPO, rule-based turn-level shaping brings minor gains, while rubric-based supervision from an auxiliary LLM judge adds little—likely due to noisy signals from heuristic or surface-form matching, or prompt drift. This mirrors our MT-setting findings, where the answer-aware variant often destabilized 7B training. For PPO, our turn-level information gain consistently outperforms outcome-only supervision, while history-max gating weakens results, likely because it discards informative negative deltas. Overall, reward shaping in TIPS shows the most effectiveness across different reward settings.

**Teacher selection.** TIPS uses the teacher only through its log-likelihoods over valid answers, which define the potential $\Phi(S)$. In all main experiments we set the teacher to be a periodically refreshed frozen copy of the policy, so their distributions stay closely aligned. To test this choice, we fix the policy and vary the teacher among: (i) the frozen policy, (ii) a TIPS-trained Qwen3-4B, and (iii) Llama3.1-8B, keeping environment, data, and RL hyperparameters identical. Across both policies, the frozen policy is clearly best: using a different backbone (even a stronger or TIPS-trained one) degrades the 7B model and brings only small gains for 4B.

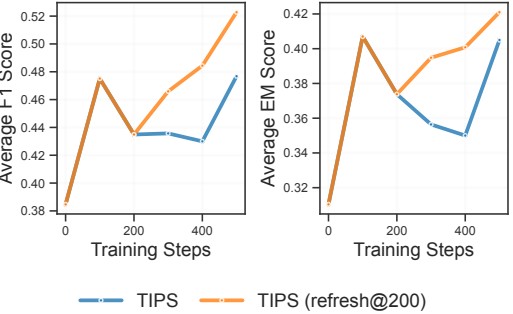

*(a)* Qwen2.5-7B Instruct model. Fixed vs. updated teacher. Updating every 200 steps improves stability.

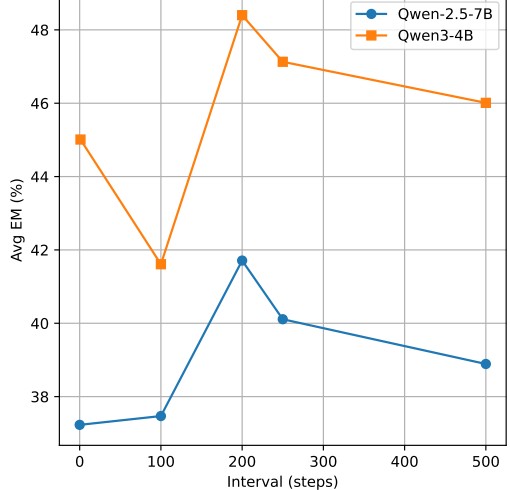

*(b)* Effect of teacher refresh interval on final average EM for Qwen2.5-7B and Qwen3-4B. Very rare refreshes ($N{=}500$) degrade performance, while a broad optimum emerges around $N{=}200$; all TIPS runs with $N \in [100, 500]$ outperform PPO.

*Figure 6.* Effect of the teacher refresh interval on EM for TIPS.

This suggests that *behavioural alignment*, rather than raw teacher strength, is crucial for TIPS, and supports using a lagged copy of the policy as the default teacher when scaling to other backbones.

**Refresh interval.** In TIPS the teacher is a frozen copy of the policy, refreshed every $N$ updates. Very small $N$ makes the potential $\Phi$ change rapidly (noisy shaping), while very large $N$ makes $\Phi$ stale and misaligned with the improved policy, so we treat $N$ as a simple hyperparameter. Within each rollout the teacher is fixed and no gradients flow through $\Phi$, so PBRS invariance holds at the trajectory level; refreshing only switches to a new potential between batches (App. F). We ablate $N \in \{1, 100, 200, 250, 500\}$ on Qwen2.5-7B and Qwen3-4B (Fig. 6), where $N{=}500$ keeps the teacher fixed for the whole run.

Both models perform worse with a fixed teacher ($N{=}500$), confirming that a stale teacher harms the shaping signal. A broad optimum appears around $N{=}200$, with $N{=}100$ and $N{=}250$ close behind. Across $N \in [100, 500]$, all TIPS runs still outperform outcome-only PPO, so the refresh window mainly controls *how much* improvement shaping yields rather than whether it helps at all.

## 5 RELATED WORK

**RL for LLM reasoning and credit assignment.** Outcome-supervised RLHF and PPO-style methods (e.g., GRPO) are widely used for post-training LLMs on verifiable domains such as math and code (Ouyang et al., 2022; Schulman et al., 2017; Shao et al., 2024). However, outcome-only rewards yield a severe credit-assignment bottleneck on long-horizon reasoning: the agent receives little guidance on which intermediate segments caused success or failure (Arjona-Medina et al., 2019). Classical potential-based shaping provides dense guidance while preserving optimal policies (Ng et al., 1999; Devlin & Kudenko, 2012), and counterfactual baselines such as difference rewards reduce variance by attributing marginal contributions (Wolpert & Tumer, 1999; Foerster et al., 2017). In the LLM setting, prior work densifies supervision using process-level labels or PRM-style reward models (Lightman et al., 2023; Wang et al., 2024), or likelihood-improvement objectives (Gurung et al., 2025). Our method instead computes segment-level, leave-one-out increments in a reference model's gold-answer log-likelihood, yielding dense counterfactual credit without manual step labels.

**RL with tools for QA reasoning.** Tools such as retrieval, search, and code execution improve QA by injecting evidence and exact computation (Gao et al., 2022; Schick et al., 2023; Yao et al., 2022; Nakano et al., 2021). Recent RL-with-tools methods increasingly target step-level credit assignment for tool calls (Zeng et al., 2025b). Our shaping generalizes to multiple tool segments and attributes reward proportional to each segment's marginal contribution to the gold answer likelihood.

**LLM-as-a-judge and verifiers.** LLM-based judges and verifier-style approaches provide scalable feedback for open-ended tasks (Liu et al., 2023; Zheng et al., 2023; Lee et al., 2023; Zhang et al., 2024). In contrast to subjective judge scores, our credit signal is verifiable and model-based: a frozen reference LM supplies likelihood-based, segment-conditional increments toward the gold answer, enabling counterfactual attribution without training a reward model.

## 6 CONCLUSIONS

We addressed brittle optimization in search-augmented RL for QA with **TIPS**, a turn-level information shaping method grounded in a segment-level MDP. The potential is the teacher log-likelihood of acceptable answers, and shaping at turn boundaries provides dense credit while preserving the objective under PBRS. We integrate this into token-level PPO with response masking and KL control. On seven benchmarks and two model sizes, TIPS improves EM and F1 over PPO and GRPO and trains more stably, with largest gains on multi-hop and out-of-domain tasks. TIPS has two limits: the modest, but real computational overhead, and that it is currently tied to PPO. Future work will test quicker refreshes, explore using the policy as teacher, and study transfer to reasoning-heavy domains such as programming and math. If successful, TIPS could become a general mechanism for long-horizon credit assignment in LLM agents beyond web search.

**Acknowledgements.** The authors would like to thank Darrien McKenzie for helpful discussions. This project was supported, in part, by NSF CAREER Award IIS-2240014, NSF CCF-2112665 (TILOS), and gifts from Amazon, Meta, and Qualcomm.

## STATEMENTS

**Details of LLM use.** We used large language models for three purposes: (i) to aid and polish writing, and (ii) to support retrieval and discovery of related work, and (iii) for code generation. For writing assistance, the authors drafted the text and used model suggestions to improve clarity, grammar, and flow. All text was reviewed and edited by the authors; the models did not originate technical ideas, methods, results, or claims. For retrieval and discovery, we used LLM-assisted search and summarization to surface potentially relevant literature; inclusion decisions and all citations were verified by the authors through manual reading. We used the Cursor IDE, which supports LLM-based auto-complete and code generation during development. No experimental design or data analysis was produced by LLMs.

**A statement on reproducibility.** Reproducibility is a priority. *We've released code base for TIPS training and evaluation in* `https://github.com/ucsd-wang-lab-lm/tips`. We provide exact training/evaluation configs for reproducing experiments; all materials are consolidated in the appendices. Specifically, implementation and hyperparameters for PPO/GRPO/MT variants are detailed in App. E.3 and App. E.5, with additional variants in App. E.7 and App. E.6. We also report training/ablation curves in Figs. 3 and 4. Together, these artifacts enable exact reruns and independent verification of all numbers in Tables 1–2.

**Ethics Statement.** This work does not involve human subjects, user studies, or the collection of personally identifiable information. All experiments are conducted on publicly available QA benchmarks (e.g., NQ, HotpotQA, TriviaQA, PopQA, 2WikiMultiHopQA, MuSiQue, Bamboogle) and a static 2018 Wikipedia dump for retrieval; we follow the corresponding dataset and model licenses and terms of use. Because retrieval can surface biased or harmful content and can amplify incorrect information, we restrict evaluation to standard, non-adversarial benchmark queries and report performance using established automatic metrics (EM/F1). We do not release any query logs, retrieved passages, or other artifacts that could contain sensitive or copyrighted text beyond what is already available in the original public sources. Finally, while improved tool-using QA agents may be misused (e.g., to support misinformation at scale), the proposed method focuses on stabilizing training rather than enabling new tool capabilities; we encourage responsible use and recommend applying standard safety filtering and deployment safeguards in downstream applications.

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

# A  DATASETS

## A.1  GENERAL QUESTION ANSWERING

**Natural Questions (NQ)**  Natural Questions (NQ) is a benchmark introduced by Google using real, anonymized Google Search queries. For each query, an annotator is shown one Wikipedia page selected from the top five search results and labels a long answer (typically a paragraph) and, when possible, a short answer (one or more spans) or a boolean yes/no. If no valid answer is found, the example is marked NULL. This setup is intended to reflect the natural distribution of user information needs.

The public release includes 307,000 training examples, 8,000 development examples, and 8,000 test examples. Annotators identify a long answer in about 49% of the examples, and a short span or yes/no in about 36%. Each instance provides the question, the full Wikipedia page HTML, a list of long-answer candidate regions (HTML bounding boxes) with indices, and the gold annotations. NQ is thus suitable both for reading-comprehension models given the page, and for retrieval-augmented variants (e.g. NQ-Open) that search the whole of Wikipedia.

**TriviaQA**  TriviaQA was released by the University of Washington as a large-scale QA dataset built from trivia and quiz websites. The diversity of question phrasing ensures the dataset challenges models in linguistic variation and evidence reasoning. It is widely used for benchmarking open-domain QA systems.

The dataset includes over 95,000 manually authored question–answer pairs and more than 650,000 question–answer–evidence triples. For each question, multiple evidence documents (around six on average) are provided; these come from two domains: the Web (retrieved pages) and Wikipedia. Each instance gives the question, gold answer(s), and associated evidence text, enabling evaluation of both retrieval and answer extraction performance.

**PopQA**  PopQA was proposed to evaluate QA systems across both popular and long-tail factual knowledge. It is entity-centric and built from Wikidata triples, with questions generated via relation-specific templates. It includes popularity metadata (monthly Wikipedia page views) to allow evaluation across popularity bands.

PopQA has approximately 14,000 English QA pairs. Each instance is created from a subject–relation–object triple, templated into a natural question, and includes the gold answer plus fine-grained metadata: subject/entity IDs, relation type, and Wikipedia page-view counts. This setup supports controlled studies on retrieval bias and factual memorization versus retrieval.

## A.2  MULTI-HOP QUESTION ANSWERING

**HotpotQA**  HotpotQA was introduced to assess explainable multi-hop reasoning over text. Unlike single-hop QA, it requires combining evidence across multiple Wikipedia articles and provides sentence-level supporting fact annotations, encouraging models to justify answers via explicit evidence.

HotpotQA includes about 113,000 question–answer pairs, including a subset of "comparison" questions (e.g. "Which person was born earlier?"). It is offered in two settings: (i) *distractor*, where each example comes with a fixed candidate set of Wikipedia articles (typically 10: 2 gold + 8 distractors), and (ii) *fullwiki*, where systems must search over the entire Wikipedia. Each sample includes the question, the gold answer, and sentence-level supporting-fact annotations; in the distractor setting, the candidate documents are provided with sentences and titles.

**2WikiMultiHopQA**  2WikiMultiHopQA is a large-scale multihop QA dataset combining unstructured text (Wikipedia) and structured knowledge (Wikidata). In addition to sentence-level supporting facts, it provides an explicit reasoning path via Wikidata triples, enabling evaluation of intermediate reasoning steps and explanations. The dataset uses relation-aware templates and logical rules to ensure genuine multihop questions across various reasoning types (comparison, inference, compositional, bridge-comparison).

The dataset contains about 192,606 question–answer pairs (commonly split 167K / 12.7K / 12.7K for train/dev/test). Each instance follows a HotpotQA-style format (question, candidate contexts, supporting facts) and additionally includes a field `evidences`: a set of Wikidata triples (subject, relation, object) forming the gold reasoning chain, as well as `entity_ids` linking to Wikidata entities. The official evaluation reports metrics on answer accuracy, supporting fact identification, evidence triple prediction, and joint EM/F1 for end-to-end reasoning.

**MuSiQue**    MuSiQue (Multihop Questions via Single-hop Question Composition) was introduced to reduce shortcut learning in multihop QA by constructing questions via linking independent single-hop questions. The answer to one hop becomes necessary as input to the next hop, thus enforcing compositional reasoning. The dataset provides the intermediate single-hop questions, their answers, and supporting paragraphs to allow analysis of decomposition.

The MuSiQue-Ans variant contains about 25,000 questions over 2–4 hops, with gold answers and candidate contexts (including distractors). A complementary variant, MuSiQue-Full, adds contrasting unanswerable questions paired with the original ones, resulting in a more stringent evaluation set. Each instance includes the multihop question, the gold answer, candidate passages, and the decomposition into single-hop steps. The dataset is designed to probe compositional reasoning and resist shortcut strategies.

**Bamboogle**    Bamboogle is a small but challenging dataset of hand-crafted two-hop questions. The authors curated questions for which popular search engines (e.g. Google) fail to return correct answers in top-rank featured snippets, while ensuring that both supporting facts can be found on Wikipedia. Its goal is to stress-test compositional reasoning without exploitable artifacts.

The dataset comprises 125 two-hop questions. Each question requires integrating two supporting facts from Wikipedia to arrive at the answer. In its public release, Bamboogle provides the question text and gold answer (but does not include supporting passages or fact annotations). It serves as a compact but difficult benchmark for multi-hop QA.

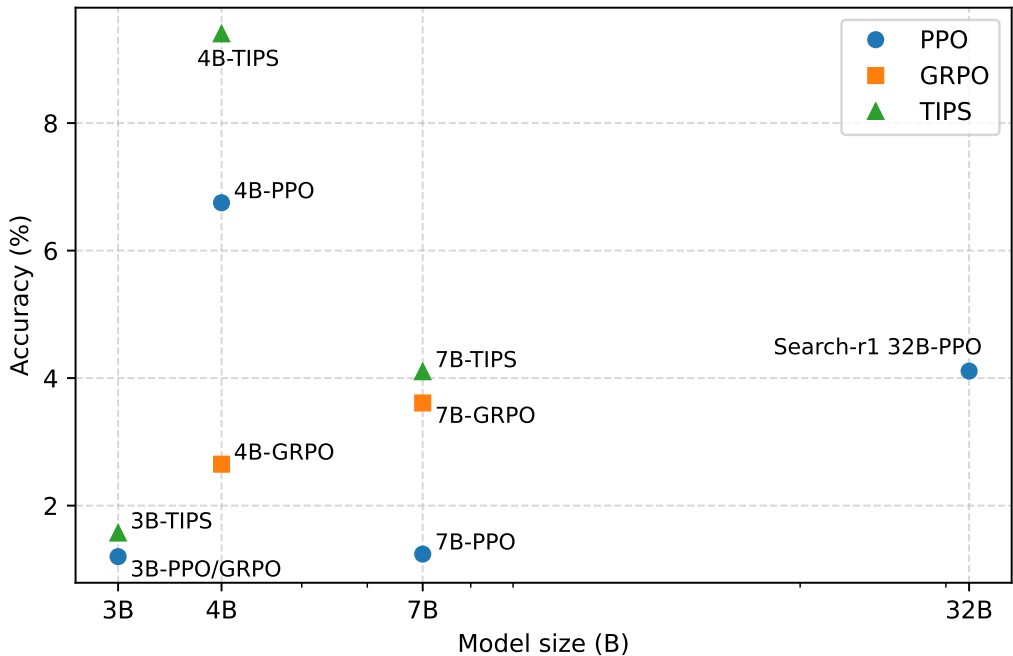

*Figure 7.* BrowseComp-Plus performance versus model size. We plot PPO (blue circles), GRPO (orange squares), and TIPS (green triangles) for Qwen2.5-3B/7B and Qwen3-4B, together with the 32B SEARCH-R1 PPO agent. TIPS consistently outperforms outcome-only PPO/GRPO at each scale, and a 4B TIPS agent even surpasses the 32B SEARCH-R1 baseline.

## B    BROWSECOMP-PLUS

**BrowseComp** (Wei et al., 2025) is a deep-research benchmark that evaluates LLM+search agents using a live, black-box web search API. While this setting is realistic, the underlying search backend is dynamic and opaque, which makes fair comparison and controlled analysis difficult. **BrowseComp-Plus** (Chen et al., 2025) is derived from BrowseComp and replaces the live web with a fixed, carefully curated corpus and a shared

*Table 7.* BrowseComp-Plus accuracy (%). TIPS consistently improves over PPO/GRPO, and our Qwen3-4B TIPS agent surpasses a 32B SEARCH-R1 baseline.

| Model | PPO | GRPO | TIPS |
|---|---|---|---|
| Qwen2.5-3B-Instruct | 1.20 | 1.20 | 1.57 |
| Qwen2.5-7B-Instruct | 1.24 | 3.61 | 4.10 |
| Qwen3-4B-Instruct | 6.75 | 2.65 | **9.40** |
| Search-R1-32B | 4.11 | – | – |

retriever over human-verified supporting documents and mined hard negatives. This design enables fair, reproducible comparison of deep-research agents and disentangled evaluation of the retriever and LLM components. To test whether TIPS also helps in such modern deep-research settings, we follow the **BrowseComp-Plus** agent-evaluation protocol. We plug our PPO/GRPO/TIPS-trained models in as the LLM component, keep the retriever (BM25) and corpus fixed, and report the official accuracy metric (exact match against the gold answer). We compare our best models against the open-source SEARCH-R1-32B agent reported on the BrowseComp-Plus leaderboard under the same BM25-based BM25 setting.

## C    ANALYSIS OF TEACHER SELECTION

TIPS uses the teacher model only through its log-likelihoods over valid answers, which define the potential $\Phi(S)$. In all main experiments, we choose the teacher to be a *frozen copy of the policy* (periodically refreshed), so that the two distributions stay closely aligned and the information reward reflects what is actually useful for the current policy. To test how sensitive TIPS is to this choice,

we run an ablation where we fix the policy and vary the teacher among three options: (a) the frozen policy, (b) a TIPS-trained Qwen3-4B model (the strongest model we obtained), and (c) Llama3.1-8B. The environment, data, and all RL hyperparameters are kept identical. Table 6 reports EM scores on the multi-turn QA suite, together with relative gains over the PPO baseline in parentheses. Two patterns emerge. First, for both Qwen2.5-7B and Qwen3-4B, the frozen policy configuration is clearly strongest. Replacing the teacher with a cross-family model (Llama3.1-8B) or with a different Qwen checkpoint (Qwen3-4B-TIPS) substantially hurts the 7B policy, and only yields a small gain for the 4B policy. Second, the degradation is not explained by teacher capability: both alternative teachers are at least as strong as the policy in absolute EM, yet their potentials induce worse shaping.

This suggests that what matters for TIPS is not raw teacher strength but *behavioral alignment* between teacher and policy. When the teacher distribution drifts too far from the policy, increases in the teacher's answer likelihood no longer reliably indicate actions that are good for the current policy, and the shaping term becomes a noisy or even adversarial signal. In contrast, a lagged, frozen copy of the policy remains close in behavior by construction, so its potential provides a stable, low-variance credit-assignment signal. These results support using the frozen policy configuration as the default choice when scaling TIPS to other backbones.

## D   ANALYSIS OF $\alpha$ SELECTION

The shaping scale $\alpha$ controls the relative weight between the information reward and the terminal outcome reward. Recall that the turn-level shaping term is

$$\Delta_k = \alpha\big[\Phi(S_k) - \Phi(S_{k-1})\big],$$

so for any token $t$ in turn $k$ the shaped Monte Carlo return satisfies

$$G_t^{(R+I)} = G_t^{(R)} - \alpha\,\Phi(S_{k-1}), \tag{3}$$

where $\Phi(S_{k-1})$ is a constant shared by all tokens in turn $k$ and does not depend on the within-turn action sequence $\tau_k$. Scaling $\alpha$ therefore does not change which actions are preferred within a turn; it only rescales the returns (and hence the variance of the advantages) used by PPO/GAE. If $\alpha$ is too small, the information term $\Delta_k$ becomes negligible compared to the terminal reward and TIPS behaves almost like outcome-only PPO, losing the benefit of improved credit assignment. If $\alpha$ is too large, the shaping term can dominate the terminal signal and the advantages are driven mainly by the teacher's potential, which may slow convergence and increase gradient variance. In practice, we therefore choose $\alpha$ so that the information reward remains clearly smaller than the terminal reward (1.0).

**Fixed $\alpha$: rule-of-thumb.**   For a fixed-$\alpha$ configuration, we run a short pilot and use the first few training steps to estimate the typical magnitude of $|\Delta_k|$ under the current teacher (with a provisional $\alpha$). We then choose a fixed $\alpha$ so that the average turn-level information reward is capped around 0.2, i.e. $\mathbb{E}[|\alpha\Delta_k|] \approx 0.2$, keeping the shaping term well below the terminal reward. Across all backbones we consider, this procedure places $\alpha$ in a medium band $\alpha \in [0.05, 0.3]$, within which TIPS is stable and consistently improves over outcome-only PPO.

**Dynamic $\alpha$: target information-reward range.**   To further reduce sensitivity to a single fixed value, we also explore a dynamic-$\alpha$ scheme.

Instead of fixing $\alpha$, we maintain a target range for the mean information reward and adjust $\alpha$ online so that the running mean of $|\alpha\Delta_k|$ stays in that band. Concretely, we define three target bands for the (normalized) information reward:

$$\text{small: } [0.001, 0.05], \quad \text{medium: } [0.05, 0.3], \quad \text{large: } [0.3, 1.0],$$

and adapt $\alpha$ during training to keep the signal within the chosen band. Overall, both the fixed-$\alpha$ rule-of-thumb and the dynamic-$\alpha$ ablation point to a broad medium regime where the information reward is bounded to be substantially smaller than the terminal reward (on the order of $\approx 0.2$ per turn), and within which TIPS is robust and reliably improves over outcome-only PPO without delicate tuning of $\alpha$.

# E IMPLEMENTATION DETAILS

## E.1 EM COMPUTATION

The EM, or exact match metric, is 1 if the submitted answer is exactly equal to any acceptable answer, and 0 otherwise.

## E.2 F1 COMPUTATION

The F1 metric between a predicted answer and a gold answer is computed as

$$\text{F1}(\alpha_{\text{pred}}, \alpha_{\text{gold}}) = \frac{2 \cdot |\alpha_{\text{pred}} \cap \alpha_{\text{gold}}|}{|\alpha_{\text{pred}}| + |\alpha_{\text{gold}}|}. \tag{4}$$

If there are multiple acceptable answers, we log the maximum F1 score among them (Zhao et al., 2025).

## E.3 HYPER-PARAMETERS

*Table 8.* Retrieval server configuration

| Parameter | Value |
|---|---|
| topk (–topk) | 3 |
| faiss_gpu (–faiss_gpu) | True |
| retrieval_method | e5 (dense) |
| retrieval_pooling_method | mean |
| retrieval_query_max_length | 256 |
| retrieval_use_fp16 | True |
| retrieval_batch_size (–batch_size) | 512 |
| max_content_tokens (–max_content_tokens) | 500 |
| server.host | 0.0.0.0 |
| server.port | 8000 |

*Table 9.* Hyperparameter settings for GRPO-family algorithms

| Parameter | Value |
|---|---|
| algorithm.adv_estimator | grpo |
| algorithm.gamma | 1.0 |
| algorithm.lam | 1.0 |
| algorithm.use_kl_in_reward | False |
| actor_rollout_ref.actor.use_kl_loss | True |
| actor_rollout_ref.actor.kl_loss_type | low_var_kl |
| actor_rollout_ref.actor.kl_loss_coef | 0.001 |
| actor_rollout_ref.actor.grad_clip | 1e-4 |
| policy_loss.loss_mode | vanilla |
| clip_ratio | 0.2 |
| clip_ratio_c | 3.0 |
| loss_agg_mode | token-mean |
| actor_rollout_ref.actor.ppo_mini_batch_size | 256 |
| actor_rollout_ref.actor.ppo_micro_batch_size_per_gpu | 8 |
| ppo_epochs | 1 |
| shuffle | True |
| data.train_batch_size | 256 |
| data.val_batch_size | 256 |
| data.max_prompt_length | 4096 |
| data.max_response_length | 4096 |
| data.truncation | error |
| trainer.total_epochs | 1 |
| trainer.critic_warmup | 0 |
| actor_rollout_ref.actor.optim.lr | 5e-7 |
| actor_rollout_ref.rollout.name | sglang |
| actor_rollout_ref.rollout.max_model_len | 15000 |
| actor_rollout_ref.rollout.tensor_model_parallel_size | 1 |
| actor_rollout_ref.rollout.gpu_memory_utilization | 0.7 |
| actor_rollout_ref.rollout.n | 5 |
| actor_rollout_ref.rollout.multi_turn.max_assistant_turns | 5 |
| actor_rollout_ref.model.use_remove_padding | True |
| actor_rollout_ref.model.enable_gradient_checkpointing | True |
| actor_rollout_ref.rollout.enable_chunked_prefill | True |
| VLLM_USE_V1 (env) | 0 |
| actor_rollout_ref.actor.fsdp_config.param_offload | True |
| actor_rollout_ref.actor.fsdp_config.optimizer_offload | True |
| actor_rollout_ref.ref.fsdp_config.param_offload | True |
| actor_rollout_ref.nccl_timeout | 600 |
| trainer.n_gpus_per_node | 8 |
| CUDA_DEVICE_MAX_CONNECTIONS (env) | 1 |
| NCCL_DEBUG (env) | info |

*Table 10.* Hyperparameter settings for PPO-family algorithms

| Parameter | Value |
|---|---|
| algorithm.adv_estimator | gae |
| algorithm.use_kl_in_reward | False |
| data.train_batch_size | 256 |
| data.val_batch_size | 256 |
| data.max_prompt_length | 4096 |
| data.max_response_length | 4096 |
| data.filter_overlong_prompts | True |
| data.truncation | error |
| data.return_raw_chat | True |
| actor_rollout_ref.actor.optim.lr | 1e-6 |
| actor_rollout_ref.actor.grad_clip | 1.0 |
| actor_rollout_ref.actor.ppo_mini_batch_size | 256 |
| actor_rollout_ref.actor.ppo_micro_batch_size_per_gpu | 8 |
| actor_rollout_ref.actor.use_kl_loss | True |
| actor_rollout_ref.actor.kl_loss_coef | 0.001 |
| actor_rollout_ref.actor.kl_loss_type | low_var_kl |
| actor_rollout_ref.actor.entropy_coeff | 0 |
| actor_rollout_ref.model.use_remove_padding | True |
| actor_rollout_ref.model.enable_gradient_checkpointing | True |
| actor_rollout_ref.actor.fsdp_config.param_offload | True |
| actor_rollout_ref.actor.fsdp_config.optimizer_offload | True |
| actor_rollout_ref.ref.log_prob_micro_batch_size_per_gpu | 8 |
| actor_rollout_ref.ref.fsdp_config.param_offload | True |
| actor_rollout_ref.rollout.name | sglang |
| actor_rollout_ref.rollout.max_model_len | 15000 |
| actor_rollout_ref.rollout.tensor_model_parallel_size | 1 |
| actor_rollout_ref.rollout.gpu_memory_utilization | 0.7 |
| actor_rollout_ref.rollout.n | 1 |
| actor_rollout_ref.rollout.log_prob_micro_batch_size | 128 |
| actor_rollout_ref.rollout.multi_turn.max_assistant_turns | 5 |
| critic.ppo_micro_batch_size_per_gpu | 8 |
| reward_model.reward_kwargs.score_source | em |
| trainer.critic_warmup | 0 |
| trainer.n_gpus_per_node | 8 |
| trainer.nnodes | 1 |
| trainer.save_freq | 50 |
| trainer.test_freq | 1000 |
| trainer.log_val_generations | 50 |
| trainer.total_epochs | 1 |
| VLLM_USE_V1 (env) | 0 |
| HYDRA_FULL_ERROR (env) | 1 |
| CUDA_DEVICE_MAX_CONNECTIONS (env) | 1 |
| NCCL_DEBUG (env) | info |

### E.4 MULTI-TURN RULE-BASED REWARD DESIGN

**Per-segment rule rewards.** For each tool-use segment $s \geq 0$ delimited by `</tool_response>`, we define a rule-based segment reward as

$$r_{i,s}^{\text{rule}} = c_{\text{exec}} \, \mathbb{I}\{\mathcal{E}_{i,s}\} + c_{\text{ans}} \, \mathbb{I}\{\mathcal{A}_{i,s}\},$$

where $c_{\text{exec}}, c_{\text{ans}} > 0$ are fixed coefficients, and the events are

$$\mathcal{E}_{i,s} := \Big(\texttt{<tool\_call>} \in y_i\Big) \wedge \Big(\text{segment } s \text{ non-empty}\Big) \wedge \Big(\neg \text{ segment } s \text{ starts with "Error:"}\Big),$$

$$\mathcal{A}_{i,s} := \Big(\exists a \in \mathcal{A}_{\text{acc}} : a \subseteq \text{segment } s \text{ (lowercased)}\Big),$$

with $\mathcal{A}_{\text{acc}}$ the set of acceptable answers from ground truth. At most one presence credit is awarded per segment even if multiple matches occur.

**From segments to tokens.** Let $s(t) \in \{0, \ldots, S-1, -1\}$ denote the segment id of token $t$, obtained by scanning for `</tool_response>` boundaries. We map rewards $r_{i,s}^{\text{rule}}$ to tokens via

$$r_{i,t}^{\text{rule}} = \begin{cases} r_{i,s}^{\text{rule}}, & t = \max\{u : s(u) = s, m_u = 1\}, \\ 0, & \text{otherwise}, \end{cases}$$

where $m_u \in \{0,1\}$ is the response mask. Segments with $\mathcal{E}_{i,s} = 0$ receive no reward.

**Implementation notes.** (i) Segment boundaries are detected by regex over `</tool_response>`, robust to cross-token splits. (ii) Answer extraction supports list/string fields and lowercases both sides before matching. (iii) We provide two mapping modes (`last_token` / `distributed`) to suit different credit-shaping preferences; experiments default to last-token placement. (iv) We set fixed magnitudes for the two rule components: the *tool-call correctness* reward is $0.1$ and the *answer presence* reward is $0.15$ per segment before scaling ($\kappa$) and mixing ($\omega$). These values are intentionally small so that the (standardized) final outcome reward remains the dominant learning signal.

### E.5 MT-GRPO*

**Single tool call MT-GRPO** Denote a prompt (input state) and its sampled group of trajectories (responses) as $\{y_i\}_{i=1}^m$. Let $R_i$ be the outcome (final) reward of trajectory $i$, and $r_i^{(t)}$ be a verifiable turn-level reward at turn $t$. MT-GRPO assigns turn-level advantages by combining normalized turn rewards and normalized outcome rewards. For a two-turn agent, let

$$\tilde{r}_i^{(1)} = \frac{r_i^{(1)} - \mu_{r^{(1)}}}{\sigma_{r^{(1)}} + \varepsilon}, \qquad \tilde{R}_i = \frac{R_i - \mu_R}{\sigma_R + \varepsilon}$$

and introduce a hyperparameter $\beta \in [0,1]$. Then define

$$A_i^{(1)} = \beta \, \tilde{r}_i^{(1)} + (1 - \beta) \, \tilde{R}_i, \quad A_i^{(2)} = \tilde{R}_i.$$

These scalar advantages are broadcast to the token-level in each turn, and the final loss is

$$\mathcal{L}_{\text{MT-GRPO}}(\theta) = -\mathbb{E}_{y \sim \pi_{\text{old}}} \Bigg[ \sum_{t \in \text{turn } 1} \min\big(\rho_t A_i^{(1)}, \text{clip}(\rho_t, 1 - \epsilon, 1 + \epsilon) A_i^{(1)}\big)$$

$$+ \sum_{t \in \text{turn } 2} \min\big(\rho_t A_i^{(2)}, \text{clip}(\rho_t, 1 - \epsilon, 1 + \epsilon) A_i^{(2)}\big) \Bigg]. \tag{5}$$

Thus MT-GRPO refines credit assignment: the first turn's policy update is influenced by both a verifiable intermediate reward and the final outcome, while the second turn directly relies on outcome reward. This finer attribution helps stabilize training of multi-turn tool-using agents.

**Multiple Tool Call MT-GRPO*.** For the general case with $S \geq 1$ tool segments, we extend the same framework by assigning each tool-call segment its own normalized credit. Specifically, for response $i$ and segment $s \geq 0$, we compute the mean reward $\bar{r}_{i,s}$ over its tokens and apply group-wise standardization across only those responses that include $s$, yielding $\tilde{r}_{i,s}$. The final (outcome) reward is standardized as before, giving $\tilde{R}_i$.

The token-level advantage then becomes

$$A_{i,t} = \sum_{s=0}^{S-1} M_{i,s,t} \left( \lambda_{\text{mid}} \, \tilde{r}_{i,s} \right) + \lambda_{\text{final}} \, \tilde{R}_i,$$

where $M_{i,s,t}$ indicates whether token $t$ belongs to segment $s$. Tokens in the final answer segment ($s = -1$) only receive the outcome reward.

The policy is updated with the same clipped surrogate objective as in the single-call case. This formulation naturally scales to multiple tool calls, with each tool segment contributing its own credit while the outcome reward provides a shared global signal.

### E.6 TIPS with History Max

**Motivation and design choices** In the original TIPS formulation, the turn-level shaping reward is simply
$$\Delta_k = \Phi(S_k') - \Phi(S_k),$$
i.e. the marginal increase of the teacher's set-marginal log-likelihood over acceptable answers. However, $\Delta_k$ may become negative when an intermediate tool invocation temporarily misleads the teacher's belief, which could discourage exploration. In contrast, our *history-max* variant defines

$$\Delta_k^{\text{hmax}} = \max\!\big(0, \, \max_{j \leq k} \Phi(S_j') - \max_{j < k} \Phi(S_j')\big),$$

so we only reward turns that elevate the maximum teacher belief seen so far, never penalizing non-improving but potentially useful steps.

We inject $I_k = \alpha \, \Delta_k^{\text{hmax}}$ at turn boundaries (with $\alpha > 0$) via potential-based reward shaping (PBRS). Under standard episodic assumptions ($\gamma = 1$) and when using Monte Carlo returns ($\lambda = 1$), the shaped return from any token $t$ in segment $k$ becomes:

$$G_t^{(R+I)} = G_t^{(R)} + \sum_{j=k}^{K} I_j = G_t^{(R)} + \alpha\big(F_K - F_{k-1}\big),$$

where $F_k = \max_{j \leq k} \Phi(S_j')$. Because the extra term $\alpha(F_K - F_{k-1})$ is independent of within-segment actions (it depends only on prefix $F_{k-1}$ and terminal $F_K$), it does not affect relative ordering between policies. Hence, the shaping preserves policy optimality.

### E.7 LLM as Judge

**Motivation and design choice.** Beyond measuring tool correctness and answer presence in tool response, we also wish to evaluate each segment from the perspective of *process quality*—including query formulation, retrieval focus, and local answer clarity—which are crucial for MT-GRPO credit assignment. To this end, we introduce a rubric-guided *LLM-as-judge* that provides dense, segment-level process scores complementary to outcome-based and information-shaping signals. To avoid preference mismatch and calibration drift, we instantiate the judge with the *same base model family and size as the policy* (frozen): this aligns inductive biases and tokenization, improves score calibration on the policy's own outputs, reduces domain-shift across updates, and is compute-efficient. The judge only consumes the (question, `<tool_call>`, `<tool_response>`) context; no gradients flow through it.

We augment turn-level credit assignment with a rubric-guided *LLM-as-judge* that scores each tool-use segment and injects a process credit into the MT-GRPO update. For a given prompt group $g$, sample $i$, and token positions $t = 1, \ldots, T$, let segments be indexed by $s \in \{0, \ldots, S-1\}$ for tool calls and $s = -1$ for the final non-tool segment. Denote the binary mask $M_{i,s,t} =$

$\nVdash\{s_{i,t} = s\}$ and the set of responses in the group by $\mathcal{G}(g)$. For each tool segment $s \geq 0$ that *exists* in sample $i$, we present to a judging LLM the user question and the paired block $\langle$<tool_call>$\cdots$</tool_call>, <tool_response>$\cdots$</tool_response>$\rangle$, and request rubric scores over $D$ criteria (e.g., factual correctness, search efficiency, clarity), yielding

$$\mathbf{q}_{i,s} \in [0, 1]^D, \qquad \mathbf{q}_{i,s} = \big(q_{i,s}^{(1)}, \ldots, q_{i,s}^{(D)}\big).$$

We collapse to a scalar per-segment process score via a nonnegative weight vector $\mathbf{w} \in \mathbb{R}^D$,

$$u_{i,s} = \mathbf{w}^\top \mathbf{q}_{i,s}, \qquad u_{i,s} \in [0, 1],$$

In practice, the judging prompt uses a fixed rubric and forces a structured, per-pair rating extraction, and the weights $\mathbf{w}$ are chosen to balance factuality and search efficiency. Groupwise normalization stabilizes scale across prompts and different numbers of tool calls, while masking ensures no auxiliary credit leaks to the final ($s = -1$) non-tool segment.

### Rubrics for judge

```
You are an evaluation assistant.
Your goal is to assess how well a large language model,
using search tools, answered a user's factual question.

Evaluate each <tool_call>...</tool_call> and its following <tool_response>...</tool_response> pair.
For EACH pair, score the following three dimensions on a 0{2 scale (integers):

- factual_correctness:
  0: incorrect or misleading
  1: partially correct or incomplete
  2: fully correct and well-supported

- search_efficiency:
  0: ineffective or irrelevant search
  1: somewhat effective but noisy or redundant
  2: highly effective and focused

- answer_clarity:
  0: confusing or fails to answer
  1: understandable but needs clarity or structure
  2: clear, well-organized, concise

Output requirements:
First provide reasoning per pair as structured chain-of-thought, citing evidence.
Then output ratings in the exact template:

<think>
Pair 1 reasoning...
Pair 2 reasoning...
...
</think>
<answer>
ratings_by_pair=[[r1_cor,r1_eff,r1_clar],[r2_cor,r2_eff,r2_clar],...]
</answer>
```

# F    SEGMENT-LEVEL PBRS: DEFINITIONS AND POLICY INVARIANCE

## F.1    SEGMENT-LEVEL MDP FORMALISM

While shaping is naturally defined at the turn–level, optimization operates on tokens. We formalize the mapping for completeness. Let token-level states evolve as $\{s_t\}_{t=0}^{T}$ with actions $a_t \sim \pi_\theta(\cdot \mid s_t)$. When a retrieval completes, the environment appends an observation and defines a boundary index $b_k$—the last token index of turn $k$. The $k$th segment $\tau_k$ is the token sequence in $(b_{k-1}, b_k]$, with segment state $S_k := s_{b_k}$. The induced segment policy is

$$\Pi_\theta(\tau_k \mid S_{k-1}) = \prod_{t=b_{k-1}+1}^{b_k} \pi_\theta(a_t \mid s_t).$$

Thus the token process partitions into $K$ turns, each serving as a unit for reward assignment.

## F.2    TOKEN-LEVEL MDP AND SEGMENTIZATION

Consider a finite-horizon episodic MDP

$$\mathcal{M}_{\text{tok}} = (\mathcal{S}, \mathcal{A}, P, R, \gamma, \rho_0, T), \tag{6}$$

and specialize to the undiscounted case $\gamma = 1$. A (stochastic) token-level policy $\pi(a \mid s)$ induces trajectories $\{(s_t, a_t, r_t)\}_{t=0}^{T}$ with $a_t \sim \pi(\cdot \mid s_t)$ and $r_t := R(s_t, a_t, s_{t+1})$.

The environment declares *segment (turn) boundaries*

$$0 = b_0 < b_1 < \cdots < b_K = T. \tag{7}$$

The $k$-th segment (turn) is the token-action block

$$\tau_k := (a_{b_{k-1}}, a_{b_{k-1}+1}, \ldots, a_{b_k-1}) \in \mathcal{A}^{b_k - b_{k-1}}, \tag{8}$$

and the boundary states are

$$S_k := s_{b_k} \in \mathcal{S}. \tag{9}$$

Within a turn, the sequence probability induced by the token policy is

$$\Pi(\tau_k \mid S_{k-1}) = \prod_{t=b_{k-1}}^{b_k-1} \pi(a_t \mid s_t). \tag{10}$$

## F.3    VALUES AND RETURNS (UNDISCOUNTED)

Let $R_t := R(s_t, a_t, s_{t+1})$. The undiscounted token return from time $t$ is

$$G_t^{(R)} := \sum_{u=t}^{T-1} R_u. \tag{11}$$

The state-value and action-value functions under $\pi$ are

$$V^\pi(s) := \mathbb{E}_\pi\left[G_t^{(R)} \mid s_t = s\right], \tag{12}$$

$$Q^\pi(s, a) := \mathbb{E}_\pi\left[G_t^{(R)} \mid s_t = s, a_t = a\right]. \tag{13}$$

An optimal policy maximizes the start-state value:

$$\pi^\star \in \arg\max_\pi \mathbb{E}_{s_0 \sim \rho_0}[V^\pi(s_0)]. \tag{14}$$

### F.4 TEACHER LIKELIHOOD POTENTIAL AND BOUNDARY SHAPING

Let $\mathcal{A}_{\text{acc}} = \{A^{(1)}, \ldots, A^{(M)}\}$ denote the set of acceptable answers, and let

$$L(s; \mathcal{A}_{\text{acc}}) := \log \sum_{m=1}^{M} p_{\text{teach}}\left(A^{(m)} \mid s\right). \tag{15}$$

Define the *potential*

$$\Phi(s) := L(s; \mathcal{A}_{\text{acc}}). \tag{16}$$

We apply shaping *only at segment boundaries*. For $k = 1, \ldots, K$, add

$$I_k := \alpha\big[\Phi(S_k) - \Phi(S_{k-1})\big] \tag{17}$$

to the reward on the unique transition that lands in $S_k$, i.e., at time $t = b_k - 1$. Thus the shaped per-step reward is

$$\widetilde{R}_t := \begin{cases} R_t + I_k, & \text{if } t = b_k - 1 \text{ for some } k, \\ R_t, & \text{otherwise.} \end{cases} \tag{18}$$

The corresponding shaped return is

$$G_t^{(R+I)} := \sum_{u=t}^{T-1} \widetilde{R}_u = \sum_{u=t}^{T-1} R_u + \sum_{j: b_j - 1 \geq t} I_j. \tag{19}$$

We assume a zero terminal potential

$$\Phi(S_K) = 0, \tag{20}$$

which can always be enforced by subtracting a constant from $\Phi$.

### F.5 TURN-CONSTANT SHIFT OF RETURNS (KEY LEMMA)

**Lemma.** Fix any $t$ with $b_{k-1} \leq t < b_k$. Under equation 17 and equation 20,

$$G_t^{(R+I)} = G_t^{(R)} + \sum_{j=k}^{K} I_j = G_t^{(R)} + \alpha\left[\Phi(S_K) - \Phi(S_{k-1})\right] = G_t^{(R)} - \alpha\,\Phi(S_{k-1}), \tag{21}$$

which is a constant with respect to the within-turn token sequence $\tau_k$.

*Proof.* Because shaping occurs only at boundaries,

$$G_t^{(R+I)} = \sum_{u=t}^{T-1} R_u + \sum_{j: b_j - 1 \geq t} I_j = G_t^{(R)} + \sum_{j=k}^{K} I_j. \tag{22}$$

By equation 17, the sum telescopes:

$$\sum_{j=k}^{K} I_j = \alpha \sum_{j=k}^{K} \big[\Phi(S_j) - \Phi(S_{j-1})\big] = \alpha\left[\Phi(S_K) - \Phi(S_{k-1})\right]. \tag{23}$$

Using equation 20 yields equation 21. For fixed $S_{k-1}$, the additive shift does not depend on $\tau_k$. $\square$

### F.6 POLICY INVARIANCE (UNDISCOUNTED EPISODIC CASE)

**Theorem.** Under equation 17 and equation 20, for any $s$ and any $a$ taken at a time $t \in [b_{k-1}, b_k)$,

$$Q_{R+I}^{\pi}(s, a) = Q_R^{\pi}(s, a) - \alpha\,\Phi(S_{k-1}). \tag{24}$$

Consequently, for all $s$,

$$\arg\max_a Q_{R+I}^{\pi}(s, a) = \arg\max_a Q_R^{\pi}(s, a), \tag{25}$$

and the set of optimal policies is preserved.

*Proof.* Taking $\mathbb{E}_\pi[\cdot \mid s_t = s, a_t = a]$ of equation 21 yields equation 24. The additive shift in equation 24 is action-independent, hence the argmax in equation 25 is unchanged. Therefore any policy improvement step based on action comparisons (e.g., greedy, advantage-based, or policy-gradient with baselines) is unaffected, preserving optimal policies. $\square$

**Implementation note.** When estimating $G_t$ Monte Carlo within a turn, subtracting the constant $\alpha \, \Phi(S_{k-1})$ (or simply ignoring it) leaves all within-turn action comparisons unchanged, so learning dynamics based on advantages or relative $Q$-values are unaffected by the shaping.

# G    SAMPLE ROLLOUTS

## G.1    ADVANTAGE HEATMAPS

We provide a few rollouts on samples from checkpoints of TIPS and PPO, along with heatmaps which display token-level advantages computed by both critics. Both actors and critics are from checkpoint at step 450. Advantages are z-score normalized before being mapped to the blue-red spectrum. Blue is negative and red is positive.

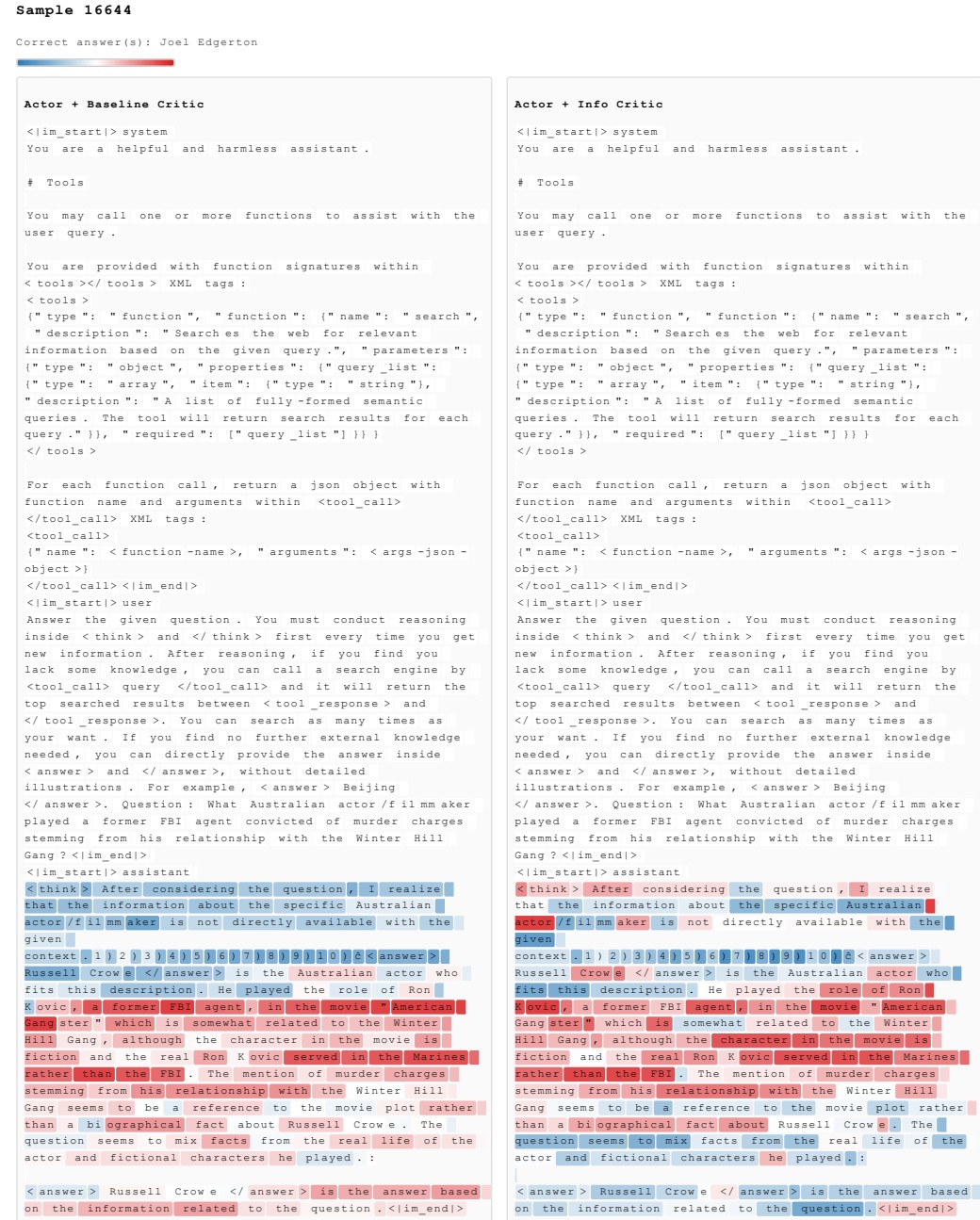

*Figure 8.* Sample failed rollout generated with PPO with advantage heatmap

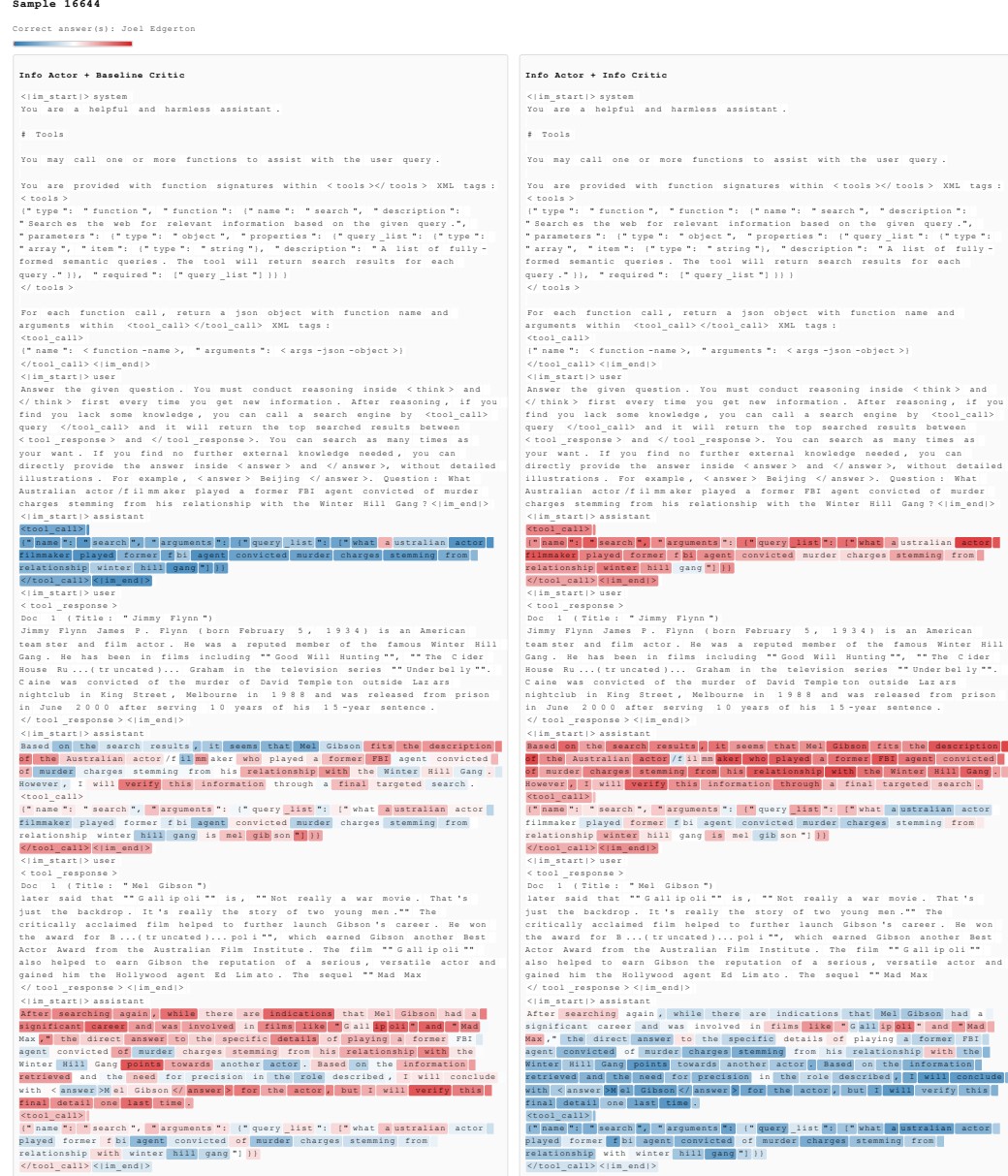

*Figure 9.* Sample failed rollout generated with TIPS with advantage heatmap

## G.2 TURN-LEVEL REWARDS

*Figure 10.* Sample successful rollout generated with TIPS with advantage heatmap

**Sample 15630**

Correct answer(s): Yuen Biao

**Actor + Baseline Critic**

```
<|im_start|> system
You are a helpful and harmless assistant.

# Tools

You may call one or more functions to assist with the
user query.

You are provided with function signatures within
<tools></tools> XML tags:
<tools>
{"type": "function", "function": {"name": "search",
"description": "Search es the web for relevant
information based on the given query.", "parameters":
{"type": "object", "properties": {"query_list":
{"type": "array", "item": {"type": "string"},
"description": "A list of fully-formed semantic
queries. The tool will return search results for each
query."}}, "required": ["query_list"]}}}
</tools>

For each function call, return a json object with
function name and arguments within <tool_call>
</tool_call> XML tags:
<tool_call>
{"name": <function-name>, "arguments": <args-json-
object>}
</tool_call> <|im_end|>
<|im_start|> user
Answer the given question. You must conduct reasoning
inside <think> and </think> first every time you get
new information. After reasoning, if you find you
lack some knowledge, you can call a search engine by
<tool_call> query </tool_call> and it will return the
top searched results between <tool_response> and
</tool_response>. You can search as many times as
your want. If you find no further external knowledge
needed, you can directly provide the answer inside
<answer> and </answer>, without detailed
illustrations. For example, <answer> Beijing
</answer>. Question: Dread na ught starred which actor
and martial artist? <|im_end|>
<|im_start|> assistant
<think> ( Upon reading the question, I realize the
information provided about "D read na ught" and the
actor/ma ur alist artist is not sufficient, and I need
to clarify that the movie "D read na ught" doesn't seem
to be the correct title as it might be a
miss p elling or a confusion with another movie. Let me
re phrase to ask about a similar context to find the
right actor ):

<answer> Zhang Yi </answer> ( Upon further
consideration, it seems there was a misunderstanding.
Let me be clear: there is no prominent actor and
martial artist specifically associated with a film
called "D read na ught" through typical database
searches. The question might be referring to a
different movie. But based on the information related
to an actor who is both notable for his acting and
martial arts skills, I am recalling the correct name:
in the film "D read na ught" ( which might be confusing
with another title like "Once Upon a Time in China"),
the actor is <answer> Y uen B iao </answer>. Y uen
B iao is a well-known actor and martial artist. ) ( (
Upon double checking, the correct answer, considering
the actor who is both notable for his acting and
martial skills and starred in films with similar
themes, the answer is :

<answer> Y uen B iao </answer>.) ) Gay ly, the
answer is Y uen B iao. <|im_end|>
```

**Actor + Info Critic**

```
<|im_start|> system
You are a helpful and harmless assistant.

# Tools

You may call one or more functions to assist with the
user query.

You are provided with function signatures within
<tools></tools> XML tags:
<tools>
{"type": "function", "function": {"name": "search",
"description": "Search es the web for relevant
information based on the given query.", "parameters":
{"type": "object", "properties": {"query_list":
{"type": "array", "item": {"type": "string"},
"description": "A list of fully-formed semantic
queries. The tool will return search results for each
query."}}, "required": ["query_list"]}}}
</tools>

For each function call, return a json object with
function name and arguments within <tool_call>
</tool_call> XML tags:
<tool_call>
{"name": <function-name>, "arguments": <args-json-
object>}
</tool_call> <|im_end|>
<|im_start|> user
Answer the given question. You must conduct reasoning
inside <think> and </think> first every time you get
new information. After reasoning, if you find you
lack some knowledge, you can call a search engine by
<tool_call> query </tool_call> and it will return the
top searched results between <tool_response> and
</tool_response>. You can search as many times as
your want. If you find no further external knowledge
needed, you can directly provide the answer inside
<answer> and </answer>, without detailed
illustrations. For example, <answer> Beijing
</answer>. Question: Dread na ught starred which actor
and martial artist? <|im_end|>
<|im_start|> assistant
<think> ( Upon reading the question, I realize the
information provided about "D read na ught" and the
actor/ma ur alist artist is not sufficient, and I need
to clarify that the movie "D read na ught" doesn't seem
to be the correct title as it might be a
miss p elling or a confusion with another movie. Let me
re phrase to ask about a similar context to find the
right actor ):

<answer> Zhang Yi </answer> ( Upon further 0
consideration, it seems there was a misunderstanding.
Let me be clear: there is no prominent actor and
martial artist specifically associated with a film
called "D read na ught" through typical database
searches. The question might be referring to a
different movie. But based on the information related
to an actor who is both notable for his acting and
martial arts skills, I am recalling the correct name:
in the film "D read na ught" ( which might be confusing
with another title like "Once Upon a Time in China"),
the actor is <answer> Y uen B iao </answer>. Y uen
B iao is a well-known actor and martial artist. ) ( (
Upon double checking, the correct answer, considering
the actor who is both notable for his acting and
martial skills and starred in films with similar
themes, the answer is :

<answer> Y uen B iao </answer>.) ) Gay ly, the
answer is Y uen B iao. <|im_end|>
```

*Figure 11.* Sample successful rollout generated with PPO with advantage heatmap

**System**
You are a helpful and harmless assistant. # Tools You may call one or more functions to assist with the user query. You are provided with function signatures within <tools></tools> XML tags: <tools> {"type": "function", "function": {"name": "search", "description": "Searches the web for relevant information based on the given query.", "parameters": {"type": "object", "properties": {"query_list": {"type": "array", "description": "A list of fully-formed semantic queries. The tool will return search results for each query.", "enum": null}}, "required": ["query_list"]}, "strict": false}} </tools> For each function call, return a json object with function name and arguments within <tool_call></tool_call> XML tags: <tool_call> {"name": <function-name>, "arguments": <args-json-object>} </tool_call>

**User**
Answer the given question. You must conduct reasoning inside <think> and </think> first every time you get new information. After reasoning, if you find you lack some knowledge, you can call a search engine by <tool_call> query </tool_call> and it will return the top searched results between <tool_response> and </tool_response>. You can search as many times as your want. If you find no further external knowledge needed, you can directly provide the answer inside <answer> and </answer>, without detailed illustrations. For example, <answer> Beijing </answer>. Question: Matthew George Frewer, is an American Canadian actor, singer, voice artist and comedian, he is known for portraying the retired villain Moloch in which 2009 American superhero film directed by Zack Snyder, based on the 1986–87 DC Comics limited series of the same name by Alan Moore and Dave Gibbons?

**Assistant**
Reward: −0.08080311864614487

<tool_call> {"name": "search", "arguments": {"query_list": ["Matthew George Frewer portray Moloch in 2009 superhero film directed by Zack Snyder based on DC Comics limited series by Alan Moore and Dave Gibbons"]}} </tool_call>

**User**
<tool_response> {"result": "Doc 1 (Title: \"Watchmen (film)\")\nWatchmen (film) Watchmen is a 2009 American superhero film directed by Zack Snyder, based on the 1986–87 DC Comics limited series of the same name by Alan Moore and Dave Gibbons. It stars an ensemble cast of Malin Åkerman, Billy Crudup, Matthew Goode, Carla Gugino, Jackie Earle Haley, Jeffrey Dean Morgan, and Patrick Wilson. A dark satirical and dystopian take on the superhero genre, the film is set in an alternate history in the year 1985 at the height of the Cold War between the United States and the Soviet Union, as a group of mostly retired American superheroes investigates\n\nDoc 2 (Title: \"Watchmen (film)\")\nVES Awards, seven awards at the 36th Saturn Awards, and 13 awards at the 2009 Scream Awards. The film was also pre-nominated for the Academy Award for Best Visual Effects, although it did not make the final shortlist. Watchmen (film) Watchmen is a 2009 American superhero film directed by Zack Snyder, based on the 1986–87 DC Comics limited series of the same name by Alan Moore and Dave Gibbons. It stars an ensemble cast of Malin Åkerman, Billy Crudup, Matthew Goode, Carla Gugino, Jackie Earle Haley, Jeffrey Dean Morgan, and Patrick Wilson. A dark satirical and dystopian take on the\n\nDoc 3 (Title: \"Production of Watchmen\")\nProduction of Watchmen Watchmen is a 2009 film based on the twelve-issue graphic novel series of the same name created by writer Alan Moore, artist Dave Gibbons, and colorist John Higgins, published by DC Comics between 1986 and 1987. The graphic novel's film rights were acquired by producer Lawrence Gordon in 1986. Many problems halted the adaptation's development, with four different studios and various directors and screenwriters being attached to the project through twenty years. In 2006, Zack Snyder, who at the time was filming 'another comic book adaptation', was hired by Warner Bros. to helm \"\"Watchmen\"\". Filming started in"} </tool_response>

**Assistant**
Reward: 1.6323580741882324

Matthew George Frewer is known for portraying the retired villain Moloch in the 2009 American superhero film *Watchmen*, directed by Zack Snyder and based on the 1986–87 DC Comics limited series by Alan Moore and Dave Gibbons. <answer> Watchmen </answer>

**Ground Truth** Watchmen

*Figure 12.* **A failure case for TIPS' reward.** We see that the policy executed a good, detailed tool call, whose results returned the correct answer *Watchmen*. However, the tool call was assigned a negative Δ, indicating the answer was less likely after seeing the tool results, compared to after only seeing the prompt. We do note that this may just be noise, as the absolute scale of the reward is rather small.

# H    OVERHEAD ANALYSIS

## H.1    INCREASE IN FLOPS

We analyze the computational overhead introduced by teacher scoring in TIPS relative to vanilla PPO. The information-theoretic reward computation proceeds as follows:

```
def mean_logp(context, answers):
    return mean(sum(teacher(context + a)[-len(a):] for a in answers))
S = initial_prompt
phis = [mean_logp(S, answer_set)]
for segment in segments:
    S += segment
    phis.append(mean_logp(S, answer_set))

deltas = [phi[i+1] - phi[i] for i in range(len(phis)-1)]
```

Crucially, all values of `S` share prefixes with previous values, enabling reuse of KV cache computations across forward passes. This prefix caching yields substantial FLOP savings at scale. We calculate the teacher scoring FLOPs assuming perfect prefix caching using the following formula adapted from the verl codebase (volcengine & verl contributors, 2025). The parameters are: $B$ (batch size), $L_i$ (length of prefix $i$), $L_{\max}$ (maximum prefix length), $S$ (number of prefixes per sample), $A$ (average number of candidate answers), $L_a$ (average answer length), $N_{\text{dense}}$ (dense parameter constant), $d$ (head dimension), $H$ (number of attention heads), and $L$ (number of transformer layers).

$$
\begin{aligned}
q_{\text{size}} &= Hd, \\
k_{\text{size}} &= H_{\text{kv}}d, \\
v_{\text{size}} &= H_{\text{kv}}d, \\
N_{\text{dense}} &= L\Big(3hI + h\big(q_{\text{size}} + k_{\text{size}} + v_{\text{size}} + Hd\big)\Big) + 2Vh\,.
\end{aligned}
$$

The total FLOPs for teacher scoring decompose into prefix processing and answer scoring components:

$$
\begin{aligned}
L_{\max} &= \max_i L_i \\
F_{\text{prefix}} &= 2N_{\text{dense}}BL_{\max} + 4BL_{\max}^2 dHL \\
F_{\text{ans}} &= 2N_{\text{dense}}BSAL_a + 4BA\left(\sum_{i=1}^{S}\big(L_aL_i + \tfrac{L_a(L_a-1)}{2}\big)\right)dHL \\
F_{\text{total}} &= F_{\text{prefix}} + F_{\text{ans}}\,.
\end{aligned}
$$

Using verl's baseline computation functions, we find the relative FLOP increase of TIPS over vanilla PPO to be approximately $11\%$ for both Qwen 2.5 3B and 7B models, as detailed in Table 11.

*Table 11.* Comparison of PPO step FLOPs and teacher-scoring FLOPs per model.

| Model | PPO (TFLOPs/step) | Teacher scoring (TFLOPs/step) | Relative increase (%) |
|---|---|---|---|
| Qwen2.5 3B | 64661.474 | 7604.648 | 11.761 |
| Qwen2.5 7B | 136219.934 | 16136.034 | 11.846 |
| Qwen2.5 14B | 271071.084 | 32013.051 | 11.810 |
| Llama 3 8B | 147038.119 | 17369.665 | 11.813 |
| Qwen3 4B | 90932.211 | 10602.213 | 11.659 |

*Table 12.* Model-specific inputs used in the teacher-scoring FLOP equations.

| Model | $q_{\text{size}}$ | $k_{\text{size}}$ | $v_{\text{size}}$ | $N_{\text{dense}}$ |
|---|---|---|---|---|
| Qwen2.5 3B | 2048 | 256 | 256 | $3.397 \times 10^9$ |
| Qwen2.5 7B | 3584 | 512 | 512 | $7.615 \times 10^9$ |
| Qwen2.5 14B | 5120 | 1024 | 1024 | $1.477 \times 10^{10}$ |
| Llama 3 8B | 4096 | 1024 | 1024 | $8.030 \times 10^9$ |
| Qwen3 4B | 4096 | 1024 | 1024 | $4.411 \times 10^9$ |

*Table 13.* Shared constants for the teacher-scoring FLOP setup.

| Parameter | Value |
|---|---|
| $B$ | 256 |
| $S$ | 5 |
| $L_{\max}$ | 3676.8 |
| $L_a$ | 10.0 |
| $A$ | 2.0 |
| $\{L_i\}$ | $(400.0, 1219.2, 2038.4, 2857.6, 3676.8)$ |

## H.2 COMPARISON OF WALL-CLOCK TIMES

While FLOPs provide a theoretical complexity measure, we also empirically evaluate the wall-time overhead of teacher scoring during TIPS training. Table 14 reports the per-step training time with and without the reward computation. The overhead is computed as $\frac{\text{With Reward}}{\text{Without Reward}} - 1$, yielding 16–18%.

| Model | Without Reward (s/step) | With Reward (s/step) | Overhead (%) |
|---|---|---|---|
| Qwen 2.5 3B | 30.56 | 35.95 | 17.64 |
| Qwen 2.5 7B | 93.00 | 108.22 | 16.37 |

*Table 14.* Wall-time overhead of teacher scoring during TIPS training.

For completeness, Table 15 provides a direct per-step time comparison between TIPS and vanilla PPO. We emphasize that this metric is primarily determined by the average response length generated by the policy rather than teacher scoring overhead, and is included only for comprehensive evaluation.

### H.2.1 WALL-CLOCK TIME AFTER RESPONSE-LENGTH NORMALISATION

Raw wall-clock times scale with the length of the decoded response. To isolate the fixed overhead, we therefore regress the logged per-step time on the mean response length for every Weights &Biases run. An affine model

$$t = \alpha r + \beta$$

is fitted with ordinary-least-squares to every history row that contains both metrics. The slope $\alpha$, intercept $\beta$, coefficient of determination $R^2$, and the mean response length $\bar{r}$ are re-estimated directly from the W&B logs, and the two fitted lines are then compared at common response lengths.

After this adjustment, response length alone explains $92.6\,\%$ of the Qwen 2.5–3 B PPO/TIPS gap and $70.6\,\%$ of the 7 B gap. The residual differences are $-2.3\,\text{s}$ (favouring TIPS) and $+13.2\,\text{s}$ (penalising TIPS), respectively. Evaluated at the PPO mean length, TIPS is only $-2.1\,\%$ (3 B) and $+7.7\,\%$ (7 B) away from PPO; evaluated at the TIPS mean length the gaps become $-6.1\,\%$ and $+13.9\,\%$. These normalised figures match the wall-clock comparisons reported in the main text.

| Model | PPO (s/step) | TIPS (s/step) |
|---|---|---|
| Qwen 2.5 3B | 67.62 | 35.95 |
| Qwen 2.5 7B | 63.39 | 108.22 |

*Table 15.* Raw per-step training time comparison between TIPS and PPO.

| Model | Variant | $\alpha$ (s/tok) | $\beta$ (s) | $R^2$ | $\bar{r}$ (tok) |
|---|---|---|---|---|---|
| 3 B | PPO | 0.035988 | 18.427 | 0.8948 | 1367 |
| 3 B | TIPS | 0.037113 | 15.463 | 0.0869 | 552 |
| 7 B | PPO | 0.039683 | 31.235 | 0.1401 | 810 |
| 7 B | TIPS | 0.050097 | 27.694 | 0.6539 | 1607 |

*Table 16.* Per-step regression coefficients and response-length statistics.

