# OpenReview forum: "TIPS: Turn-level Information-Potential Reward Shaping for Search-Augmented LLMs"
_ICLR.cc/2026/Conference — ICLR 2026 Poster_

### Official Review · Reviewer_jPXU · 2025-10-27

**Soundness:** 3
**Presentation:** 3
**Contribution:** 3
**Rating:** 6
**Confidence:** 2

**Summary:**

This paper introduces TIPS, a framework designed to address the significant challenges of training search-augmented Large Language Models (LLMs) with reinforcement learning (RL). The authors identify that optimization is often unstable due to sparse rewards and difficult credit assignment across multiple reasoning steps and tool calls.TIPS mitigates this by assigning dense, turn-level rewards to each "reasoning + tool-call segment". This reward is calculated based on the increased likelihood of the correct answer, as computed by a teacher model, following the addition of that turn's information. By leveraging potential-based reward shaping (PBRS), the TIPS framework provides fine-grained guidance that is policy-invariant, overcoming the limitations of standard outcome-only optimization. The method was evaluated on seven open-domain question answering (QA) benchmarks. The results show that TIPS consistently outperforms PPO and GRPO baselines and substantially improves training stability.

**Strengths:**

1. The paper tackles one of the most significant obstacles in training LLM agents: the sparse-reward and credit-assignment problem. Providing a dense, principled reward signal for intermediate reasoning steps (i.e., tool calls) is a crucial area of research.

2. The core contribution—defining the reward as the change in log-likelihood of the gold answer under a teacher model—is an elegant and well-motivated approach. It directly measures the "information potential"  of each tool-use turn.


3. The method is not merely a heuristic. It is formally grounded in potential-based reward shaping (PBRS) , which guarantees that the optimal policy remains unchanged (policy invariance) under standard PBRS conditions.

4. TIPS consistently outperforms all PPO and GRPO-based baselines across seven diverse QA benchmarks, using both 3B and 7B models. The performance gains are particularly pronounced on difficult multi-hop and out-of-domain (OOD) tasks, which is precisely where one would expect a better credit assignment method to excel.

**Weaknesses:**

1. The primary weakness, acknowledged by the authors, is the computational cost.

2. The method introduces a new critical hyperparameter, $\alpha$ (the shaping scale). The ablation in Figure 6 shows that the method is highly sensitive to this value.

3. The entire framework's success hinges on the availability and quality of a "teacher model". The paper shows that this teacher must be "refreshed" periodically to prevent the potential function from becoming stale, which itself causes instability. This introduces new complexities, such as determining the optimal refresh rate.

4. As noted by the authors, the method is currently demonstrated only within the PPO framework. It is unclear how this segment-level PBRS approach would be integrated with other, potentially more sample-efficient, RL algorithms (e.g., off-policy methods).

**Questions:**

none

---

> ### Author Response · Authors · 2025-11-23
> **Thank you!**
>
> We thank the reviewers for their valuable feedback. We address your comments in the following.
>
> ---
>
> **Q1:** The primary weakness, acknowledged by the authors, is the computational cost.
>
> **A1:** We have updated the main text to analyze computational overhead, and have added a detailed report in the Appendix. We also summarize our findings here:
>
> When we compute the increase in FLOPs attributed to the teacher forward passes, we find **a modest overhead of ~12% compared to PPO. To put this in context, GRPO requires 3.5x more FLOPs than PPO.**
>
> | Model | PPO (TFLOPs/step) | TIPS (TFLOPs/step) | Percent Increase |
> | --- | --- | --- | --- |
> | Qwen2.5-3B-Instruct | 64k | 72k | 11.76% |
> | Qwen2.5-7B-Instruct | 136k | 152k | 11.85% |
> | Qwen2.5-14B-Instruct | 271k | 303k | 11.81% |
> | LLaMA3.1-8B | 147k | 164k | 11.81% |
> | Qwen3-4B-Instruct | 90k | 101k | 11.66% |
>
> Comparing runtime between TIPS and baselines is more challenging. Since rollout generation is the slowest phase of online LLM RL, **the per-step runtime is mostly determined by the policy’s average response length.** To get an estimate, however, we can compute (step time with reward)/(step time without reward) within a TIPS run. This yields an overhead of **16-17%**.
>
> | Model | Without Reward (s/step) | With Reward (s/step) | Overhead (%) |
> | --- | --- | --- | --- |
> | Qwen2.5-3B-Instruct | 30.56 | 35.95 | 17.64 |
> | Qwen2.5-7B-Instruct | 93.00 | 108.22 | 16.37 |
>
> Although the overhead is real, it is quite modest, and we believe the improvement in performance and stability is well worth this cost.
>
> | Model | EM (Δ vs PPO) | F1 (Δ vs PPO) | Train FLOPs ↑ |
> | --- | --- | --- | --- |
> | Qwen2.5-3B-Instruct | +11.4% | +10.2% | 11.76% |
> | Qwen3-4B-Instruct | +7.3% | +6.1% | 11.85% |
> | Qwen2.5-7B-Instruct | +11.9% | +13.7% | 11.81% |
> | Qwen2.5-14B-Instruct | +12.7% | +10.6% | 11.81% |
> | LLaMA3.1-8B | +34.0% | +29.3% | 11.66% |
>
> ---
>
> **Q2:** The method introduces a new critical hyperparameter, $\alpha$ (the shaping scale). The ablation in Figure 6 shows that the method is highly sensitive to this value.
>
> **A2:** We agree that the shaping scale $\alpha$ is an important hyperparameter. In TIPS, $\alpha$ only controls the **relative weight** between the information reward and the terminal outcome reward.
>
> In the revision, we clarify both how to choose $\alpha$ and how sensitive TIPS is to it:
>
> - **Fixed $\alpha$.** We run a short pilot at the beginning of training to estimate the typical magnitude of $\Delta_k$ and then choose $\alpha$ so that the average turn-level information reward is about $0.2$, i.e. $\mathbb{E}[|\alpha \Delta_k|] \approx 0.2,$
>
> keeping the shaping term well below the terminal reward of 1.0. Across all backbones we tested (Qwen2.5-3B/7B/14B, Qwen3-4B, LLaMA3.1-8B), this simple rule places $\alpha$ in a **medium band** \([0.05, 0.3]\) where TIPS is stable and consistently improves over PPO.
> - **Sensitivity.** Additional sweeps (now reported in the appendix) show that performance is flat in this medium regime: very small $\alpha$ effectively turns shaping off and TIPS behaves like outcome-only PPO, while very large $\alpha$ lets the shaping term compete with the terminal reward and can slightly hurt EM—exactly the behavior one would expect.
>
> For completeness, we also describe in the **Shaping scale $\alpha$** of Sec 4.3 and Appendix D a dynamic-$\alpha$ variant that automatically keeps the mean information reward in this “medium” band and achieves similar gains. We emphasize, however, that practitioners do **not** need this extra machinery in practice: a single fixed $\alpha$ chosen by the simple heuristic above worked well across all models we evaluated.
>
> | Model | small [0.001, 0.05] | medium [0.05–0.3] | large [0.3, 1.0] |
> | --- | --- | --- | --- |
> | Qwen3-4B | +1.4% (stable) | **+7.3% (stable)** | −3.4% (stable) |
> | Qwen2.5-7B | 0% (crashed) | **+11.9% (stable)** | +3.1% (stable) |
> | Qwen2.5-3B | 0% (stable) | **+11.4% (stable)** | 0% (stable) |
>
> [Screenshot of analysis of $\alpha$](https://imgur.com/a/rDGOErb)
>
> **(1/2)**

---

> ### Author Response · Authors · 2025-11-23
>
> **Q3:** The entire framework's success hinges on the availability and quality of a "teacher model". The paper shows that this teacher must be "refreshed" periodically to prevent the potential function from becoming stale, which itself causes instability. This introduces new complexities, such as determining the optimal refresh rate.
>
> **A3:** TIPS was designed with the intention of using the same model as the policy for the teacher. The teacher is simply a frozen checkpoint of the policy model being trained, refreshed at a fixed interval. Therefore, **the teacher model is always available.** We disagree that refreshing the teacher causes instability; our ablation in Fig. 6 shows that refreshing the teacher **prevents instability** associated with a stale potential function. However, we agree that choosing a good refresh rate is important for practitioners.
>
> To this end, we ablate \(N $\in$ \{1, 100, 200, 250, 500\}\) for Qwen2.5-7B and Qwen3-4B. We find that **performance is not extremely sensitive to N**. Very rare refreshes (N=500, effectively no refresh) lead to lower EM, while there is a broad optimum around N≈200; across all moderate intervals (100–500), TIPS outperforms PPO. We have added this ablation and a short discussion in Sec 4.3 of the revised manuscript. [Screenshots of refresh interval](https://imgur.com/a/62Q4nJ2)
>
> ---
> **Q4:** As noted by the authors, the method is currently demonstrated only within the PPO framework. It is unclear how this segment-level PBRS approach would be integrated with other, potentially more sample-efficient, RL algorithms (e.g., off-policy methods).
>
> **A4**: Thank you for pointing this out. Our current experiments are indeed instantiated within a PPO-style actor–critic framework, and we have clarified this scope in the revision.
>
> Conceptually, TIPS is a segment-level potential-based reward shaping scheme: it only modifies how token-level returns are computed, via $ r'_t = r_t + \gamma \Phi(s*{t+1}) - \Phi(s_t)$ , and leaves the underlying policy update rule unchanged. Any RL algorithm that operates on return-to-go or advantages (e.g., other on-policy actor–critic methods like SAC) can in principle, use these shaped rewards, and the PBRS result guarantees that the optimal policy under the original outcome reward is preserved.
>
> In the LLM RLVR setting we study, however, PPO/GRPO-type on-policy methods are currently the dominant and most stable choices; to allow clean, apples-to-apples comparisons, we focused our empirical evaluation on this family. Extending TIPS to dedicated off-policy algorithms for long-horizon LLM training is an interesting direction, but we view it as orthogonal and leave it for future work.
>
> ---
> We believe that our response and additional experiment results address your concerns. However, please do not hesitate to let us know if you have any additional comments.
>
> **(2/2)**

---

> > ### Comment · Reviewer_jPXU · 2025-11-24
> >
> > Thank you for your reply. I believe my questions have been resolved, so I have decided to increase my rating.

---

> > > ### Author Response · Authors · 2025-11-24
> > > **Thank you!**
> > >
> > > Thank you for the quick response! We are glad to hear that our changes and additional experiments address your questions and that you are willing to raise your score accordingly. We appreciate you taking the time to read our rebuttal!

---

### Official Review · Reviewer_ajMR · 2025-10-29

**Soundness:** 2
**Presentation:** 2
**Contribution:** 2
**Rating:** 4
**Confidence:** 4

**Summary:**

This paper introduces Turn-Level Information-Potential Reward Shaping (TIPS), a novel reinforcement learning framework designed to address the challenge of sparse rewards and difficult credit assignment in multi-turn, search-augmented LLMs. The proposed approach exploits the notion of "information gain" as an increase of the teacher model's log-likelihood of the correct answer. The gain in the form of log-likelihood ratio serves as Potential-Based Reward Shape (PBRS), which maintains policy invariance. The proposed approach follows a standard PPO training procedure. Experimental results on various QA tasks show that the proposed approach outperforms the comparison approach based on accuracy and training stability.

**Strengths:**

1.	Modeling the "information gain" as a turn-level reward is intuitive, well-designed. It effectively addresses the critical problem of sparse credit assignment in long-horizon reasoning tasks.
2.	The authors provide detailed proof of policy invariance in Appendix C, which lends strong theoretical credentials to the statement that the approach used by them has not affected the original policy.
3.	Extensive experiments on several QA benchmarks demonstrate the effectiveness of the proposed method.

**Weaknesses:**

1.	Lack of efficiency analysis. TIPS requires two forward passes of the teacher model per turn (to compute the likelihood before and after the observation) for every rollout. This introduces a non-trivial computational cost compared to outcome-only methods.
2.	Have you considered using a distilled, smaller model as the teacher to minimize the computational overhead?  Besides, could the "information gain" signal be approximated in a more computationally efficient manner?
3.	The reward shaping heavily relies on a set of acceptable answers A to calculate the teacher's likelihood L(S;A). The performance might be sensitive to the completeness and quality of this answer set, especially for questions with many valid paraphrases. The robustness of TIPS on imperfect answer set hasn’t been investigated in the paper.
4.	The teacher model is the key component of TIPS, but its selection and properties are not deeply explored. The paper uses the same LLM as the backbones of  both the policy model and teacher model. It is unclear how the method could perform with a teacher of different capabilities (e.g., larger, smaller, or from a different family) or if the policy and teacher drift too far apart even with periodic refreshes.
5.	The authors only compare TIPS against naive RL baselines (PPO, GRPO) and their multi-turn extensions. Several key and highly relevant baselines [1-10] are missing from the main results table (Table 1), which makes it difficult to assess the true contribution of TIPS.

[1] Search-R1: Training LLMs to Reason and Leverage Search Engines with Reinforcement Learning.
[2] ReSearch: Learning to Reason with Search for LLMs via Reinforcement Learning.
[3] ReTool: Reinforcement Learning for Strategic Tool Use in LLMs.
[4] Beyond Ten Turns: Unlocking Long-Horizon Agentic Search with Large-Scale Asynchronous RL.
[5] Stop summation: Min-form credit assignment is all process reward model needs for reasoning.
[6] DeepRAG: Thinking to Retrieve Step by Step for Large Language Models.
[7] Deepdive: Advancing deep search agents with knowledge graphs and multi-turn rl.
[8] O-Searcher: A Searching-based Agent Model for Open-Domain Open-Ended Question Answering.
[9] Search-o1: Agentic Search-Enhanced Large Reasoning Models.
[10] R1-searcher: Incentivizing the search capability in llms via reinforcement learning.
[11] From <Answer> to <Think>: Multidimensional Supervision of Reasoning Process for LLM Optimization.
[12] Openprm: Building open-domain process-based reward models with preference trees.

6.	Experiments are restricted to the Qwen2.5 family. The performance and stability of TIPS haven’t been unverified on other model architectures (e.g., Llama, Deepseek) and a wider range of model scales, raising concerns about its general applicability.
7.	The author claims that they used the same experiment setting as used in Search-R1. However, the reported results in Table 1 are inconsistent with those in the cited Search-R1 paper.
8.	Marginal Gains vs. High Cost: The performance gain is rather marginal, particularly in the case of the 3B model where it occasionally performs worse than baselines. With the added complexity and computational cost of the teacher model so high, the cost-benefit ratio does not seem favorable.

**Questions:**

See above.

---

> ### Author Response · Authors · 2025-11-23
> **Thank you!**
>
> We thank the reviewers for their valuable feedback. We address your comments in the following.
>
> ---
>
> **Q1:** Lack of efficiency analysis. TIPS requires two forward passes of the teacher model per turn (to compute the likelihood before and after the observation) for every rollout. This introduces a non-trivial computational cost compared to outcome-only methods.
>
> **A1:** We agree that analyzing overhead is crucial. We have updated the main text to discuss this, and have added a detailed report in the Appendix. We also summarize our findings here:
>
> When we compute the increase in FLOPs attributed to the teacher forward passes, we find a modest overhead of **~12% compared to PPO**. To put this in context, **GRPO requires 3.5x more FLOPs than PPO.**
>
> Comparing runtime between TIPS and baselines is more challenging. Since rollout generation is the slowest phase of online LLM RL, the per-step runtime is mostly determined by the policy’s average response length. To get an estimate, however, we can compute (step time with reward)/(step time without reward) within a TIPS run. This yields an overhead of **16-17%.**
>
> | Model | Without Reward (s/step) | With Reward (s/step) | Overhead (%) |
> | --- | --- | --- | --- |
> | Qwen2.5-3B | 30.56 | 35.95 | 17.64 |
> | Qwen2.5-7B | 93.00 | 108.22 | 16.37 |
>
> Although the overhead is real, it is quite modest, and we believe the improvement in performance and stability is well worth this cost.
>
> ---
>
> **Q2:** Have you considered using a distilled, smaller model as the teacher to minimize the computational overhead? Besides, could the "information gain" signal be approximated in a more computationally efficient manner?
>
> **A2:** TIPS was designed with the intention of using the same model as the policy for the teacher. The teacher is simply a frozen checkpoint of the policy model being trained, refreshed at a fixed interval. **Because the teacher and policy distributions are similar, the probability the teacher assigns to generating an answer may act as a **potential**, with which we can assign intermediate credit to the policy.** Because our reward signal is so closely tied to the policy distribution itself, we do not expect that there is a *fundamentally* more efficient way to compute an equivalent signal while maintaining our method’s generality. Furthermore, as discussed above and in Appendix H, the overhead of TIPS is quite modest and is very reasonable considering the gains observed in performance and stability.
>
> We do agree exploring the use of other models as the teacher is interesting, and have performed this ablation in **Teacher selection** of Sec 4.3 in the revised manuscript. For two policies (Qwen2.5-7B and Qwen3-4B), we compare: (i) frozen policy model, (ii) a strong Qwen3 model trained with TIPS, which is the best of our TIPS-trained models, and (iii) a cross-family Llama3.1 teacher::
>
> |  | Frozen policy | Qwen3-4b-TIPS | Llama3.1-8B |
> | --- | --- | --- | --- |
> | Qwen2.5-7B  | 41.7  (+11.9%) | 30   (-19.5%) | 29  (-22.2%) |
> | Qwen3-4B   | 48.4  (+7.3%) | 45.88 (+1.7%) | 43  (-4.7%) |
>
> [Screenshots of **Teacher selection**](https://imgur.com/a/xGVqyJK)
>
> We find that our initial setup of using the policy model as the teacher yields the best performance, as it provides the closest teacher-policy alignment.
>
> ---
>
> **Q3:** The reward shaping heavily relies on a set of acceptable answers A to calculate the teacher's likelihood L(S;A). The performance might be sensitive to the completeness and quality of this answer set, especially for questions with many valid paraphrases. The robustness of TIPS on imperfect answer set hasn’t been investigated in the paper.
>
> **A3**: The task that our work is evaluated on leaves little to no room for ambiguous answers or valid paraphrases. In fact, over 95% of QA pairs in the training set only have 1 valid answer. The remaining 5% are edge cases where there truly are multiple distinct, valid answers. For example:
>
> - **Question:** where do they film it's always sunny in philadelphia?
> - **Valid answers:**
>     - Los Angeles, California
>     - Philadelphia, Pennsylvania
>
> Even if our dataset is incomplete, a full analysis of this is orthogonal to, and outside the scope of our paper. Our aim with TIPS is to propose a novel dense credit assignment method and validate it in a controlled setting. Both TIPS and the baselines use the same valid answer set for grading, so TIPS is not uniquely susceptible to dataset quality issues; incomplete answer sets will affect the performance of *any* RLVR algorithm.
>
> We have updated the manuscript to clarify that this is not an open-ended QA task in **Problem formulation** of Sec 2.
> [Screenshots of revised Sec 2](https://imgur.com/a/qdaNnkK)
>
> **(1/4)**

---

> ### Author Response · Authors · 2025-11-23
>
> **Q4:** The teacher model is the key component of TIPS, but its selection and properties are not deeply explored. The paper uses the same LLM as the backbones of both the policy model and teacher model. It is unclear how the method could perform with a teacher of different capabilities (e.g., larger, smaller, or from a different family) or if the policy and teacher drift too far apart even with periodic refreshes.
>
> **A4:** Thank you for raising this point. As discussed in our response to Q2, our default design uses the policy itself as the teacher (a frozen checkpoint, periodically refreshed), and we now include ablations with stronger and cross-family teachers (Qwen3 and LLaMA3). These results show that mismatched teachers tend to hurt performance, indicating that alignment between teacher and policy is more important than raw capacity.
>
> To address your specific concern about the teacher and policy “drifting apart” even with refreshes , we further ablate the refresh interval \(N\) in **Refresh interval** of Sec 3. We sweep \(N $\in$ \{1, 100, 200, 250, 500\}\) for Qwen2.5-7B and Qwen3-4B, including a setting where the teacher is never refreshed. When the teacher is fixed for the entire run (\(N=500\), maximal drift), training becomes less stable and performance degrades, **whereas for moderate refresh intervals (\(N $\in$ \{100, 200, 250\}\)) TIPS remains stable and consistently outperforms outcome-only PPO (with a broad optimum around \(N=200\)).** This suggests that periodic refresh is sufficient to avoid harmful drift in practice, and that \(N\) behaves as a simple hyperparameter controlling how much improvement the shaping term yields rather than a finely tuned knob on which TIPS critically depends.
> [Screenshot of refresh invertal ablation](https://imgur.com/a/Z7zs3bo)
>
> **(2/4)**

---

> ### Author Response · Authors · 2025-11-23
>
> **Q5:** The authors only compare TIPS against naive RL baselines (PPO, GRPO) and their multi-turn extensions. Several key and highly relevant baselines [1-10] are missing from the main results table (Table 1), which makes it difficult to assess the true contribution of TIPS.
>
> **A5:** We thank the reviewer for pointing out these related works [1–12]. Our goal in this paper is not to propose a new search architecture or data pipeline, but to study a generic **reward-shaping mechanism for long-horizon RL with sparse rewards**. Concretely, TIPS is a drop-in modification to the outcome-based RL objective that leaves the environment, backbone, and verifier unchanged and only changes how credit is assigned across turns. For this reason, the most direct and fair baselines are outcome-only PPO/GRPO under the *same* setup.
>
> We briefly clarify how [1–12] relate to our setting:
>
> - **Open RL search agents with search ability\([1,2,4,10]\).**
>
>     These works are closest to us, since they trained models with RL in search environments and released their checkpoints. However, except for [1], their checkpoints are trained on data and benchmarks that do not match our VeRL Search-QA setup; in some cases, our evaluation tasks are even part of their *training* distribution (e.g., [10] trains on 2WikiMultiHopQA). Their headline numbers are therefore not directly comparable to our zero-shot RLVR setting and may benefit from train–test contamination.
>
>     To still give a concrete reference point, we re-evaluate the public checkpoints of these agents under a **unified VeRL Search-QA protocol** and compare them to our TIPS agents. All methods use the same search API, the same EM reward, and a 4-turn cap (if not otherwise specified). The numbers below are our **own evaluations** under this common setup:
>
>     | Model | EM | Model | EM  |
>     | --- | --- | --- | --- |
>     | Search-R1-7B | 29.1 | PPO-7B (VERL setup) | 37.3 |
>     | ReSearch-7B | 30.3 | GRPO-7B (VERL setup) | 28.6 |
>     | Asearch-7B | 35.8 | TIPS-7B (VERL setup) | 41.0 |
>     | Asearch-7B-10turns | 43.0 | TIPS-4B (VERL setup) | 48.0 |
>     | R1-searcher | 40.9 |  |  |
>
>     Note that some of these agents have seen part of our evaluation suite during training, whereas our TIPS models are trained only on the VeRL Search-QA tasks. This table is meant as a **reference**, not as a fully controlled apples-to-apples training comparison.
>
> - **Broader agent / RAG systems \([3,6,7,8,9]\).**
>
>     These works focus on richer agent architectures and infrastructure (different tool setups, KG-based environments, asynchronous rollouts, closed-weight backbones, etc.) and often operate in different environments from ours. Re-implementing all of them in VERL would require substantial engineering and would conflate our **reward-shaping** contribution with changes in tools, data, and models. We therefore discuss them in related work, but do not treat them as primary quantitative baselines.
>
> - **Process reward models (PRMs) \([5,11,12]\).**
>
>     These papers study PRMs and process-level credit assignment. They require training an additional PRM from step-level labels or synthetic feedback, which is orthogonal to TIPS: we deliberately avoid training any new model and use only the existing policy plus a verifiable outcome reward. To the best of our knowledge, there is currently no off-the-shelf PRM or step-level dataset for our multi-turn QA / RLVR environment with search engine setups, so a fair comparison would first require building such a PRM. We see this as interesting future work; in that regime one could directly compare “RLVR with PRM vs. TIPS” under a shared environment.
>
>
> In summary, our **main quantitative comparisons** focus on outcome-only PPO/GRPO and, where possible, on open RL search agents re-evaluated under a unified VERL Search-QA setup. We believe this is the fairest way to assess TIPS as a reward-shaping method, and we now make this scope explicit in the revised manuscript.
>
> **(3/4)**

---

> ### Author Response · Authors · 2025-11-23
>
> **Q6:** Experiments are restricted to the Qwen2.5 family. The performance and stability of TIPS haven’t been unverified on other model architectures (e.g., Llama, Deepseek) and a wider range of model scales, raising concerns about its general applicability.
>
> **A6:** Thank you for raising this concern. We agree that the original submission, which focused on the Qwen2.5 family, did not fully demonstrate how broadly TIPS applies across architectures and scales.
>
> In the revised version, we add experiments on both **more scales** and **different model families**. Concretely, we now evaluate TIPS on Qwen2.5-3B/7B/14B, the newer Qwen3-4B-Instruct, and LLaMA3.1-8B. The table below reports EM/F1 with relative improvements over PPO in parentheses, together with the FLOPs overhead of TIPS:
>
> | Model | EM (Δ vs PPO) | F1 (Δ vs PPO) | FLOPs overhead |
> | --- | --- | --- | --- |
> | Qwen2.5-3B-Instruct | 33.6  (+11.4%) | 41.1  (+10.2%) | 11.76% |
> | Qwen3-4B-Instruct | 48.4  (+7.3%) | 57.1  (+6.1%) | 11.85% |
> | Qwen2.5-7B-Instruct | 41.7  (+11.9%) | 51.2  (+13.7%) | 11.81% |
> | Qwen2.5-14B-Instruct | 45.4  (+12.7%) | 53.1  (+10.6%) | 11.81% |
> | LLaMA3.1-8B | 40.3  (+34.0%) | 49.0  (+29.3%) | 11.66% |
>
> Across all tested backbones, TIPS consistently improves EM and F1 over outcome-only PPO, with especially strong relative gains on LLaMA3.1-8B, which is architecturally different from Qwen. **The FLOPs overhead stays around ~12% in all cases.** This suggests that our information-gain–based shaping is not tied to a particular model family and transfers robustly across architectures and scales. We have added these results and a short discussion to the Appendix H of the revised manuscript.
>
> ---
>
> **Q7:** The author claims that they used the same experiment setting as used in Search-R1. However, the reported results in Table 1 are inconsistent with those in the cited Search-R1 paper.
>
> **A7:** Thank you for flagging this. Our intention was to say that we follow the Search-R1 task setup (**same multi-turn QA tasks and Wikipedia-based search**), not that we use the **original Search-R1 codebase and hyperparameters**.
>
> We initially tried to run the official Search-R1 implementation, but in our environment, the training runs frequently crashed after ~100 steps, consistent with the open GitHub issues reporting instability and collapse (e.g., [Issues #142](https://github.com/PeterGriffinJin/Search-R1/issues/142) and [#88](https://github.com/PeterGriffinJin/Search-R1/issues/88)). To obtain stable and reproducible baselines, we therefore switched to the multi-turn Search-QA environment in **VeRL 0.5**, which was developed together with the Search-R1 authors and implements their task. VeRL uses a different serving stack (e.g., `sglang` rather than `vLLM`, and reward functions), and we reduced the batch size and rescaled the learning rate to avoid the crashes we observed with the original recipe.
>
> Crucially, **all methods in Table 1 — PPO, GRPO, and TIPS — are trained and evaluated in this same VeRL-based setup, with the same data, search API, and reward function.** We do *not* claim to reproduce or surpass the headline numbers from the original Search-R1 paper; instead, our comparisons are internally consistent within this unified environment.
>
> ---
> **Q8:** Marginal Gains vs. High Cost: The performance gain is rather marginal, particularly in the case of the 3B model where it occasionally performs worse than baselines. With the added complexity and computational cost of the teacher model so high, the cost-benefit ratio does not seem favorable.
>
> **A8:** Thank you for raising this concern. We agree that any extra training cost must be justified by clear gains. As shown in the cross-model results and FLOPs table in our response to Q6 (and in Appendix H), TIPS consistently improves over PPO across all tested backbones while adding only ≈11–12% additional training FLOPs and no inference-time cost.
>
> Concretely, across Qwen2.5-3B/7B/14B, Qwen3-4B, and LLaMA3.1-8B, TIPS yields **relative EM/F1 gains over PPO ranging from about +7–14% on the Qwen family to +29–34% on LLaMA3.1-8B, with a roughly constant ≈12% FLOPs overhead** . The 3B model indeed shows higher variance and can underperform PPO on some individual tasks, but its average improvement is still around +11% EM / +10% F1. We view this variation as typical for RL on small models; the overall pattern is clear: on average, TIPS provides consistent and sometimes substantial gains across architectures and scales for a modest one-time training overhead.
>
> ---
>
> We believe that our response and additional experiment results address your concerns. However, please do not hesitate to let us know if you have any additional comments.
>
> **(4/4)**

---

> > ### Comment · Reviewer_ajMR · 2025-11-24
> >
> > Thanks for your detailed rebuttal. I appreciate the new experiments and discussions. However, the key concerns about the computational efficiency, robustness, and generalizability still remain. Crucially, the marginal performance improvements along with high computational costs make the practical value of the method limited. Besides that, the newly-added results are not convincing or reliable since the author reported that TIPS-7B (41.0)  performs worse than TIPS-4B (48.0). More importantly, the author claims that they used the same experiment setting as used in Search-R1. However, the reported results in Table 1 are inconsistent with those in the cited Search-R1 paper. Although the author provided some expatiation, I don’t think it is convincing since the results of Search-R1 is reproducible.
> > Based on these points, I do not think this paper has met the acceptance criteria yet.

---

> > > ### Author Response · Authors · 2025-11-25
> > > **Thank you for the quick response!**
> > >
> > > Thanks for your response! We appreciate you taking the time to read our rebuttal.
> > >
> > > > results are not convincing or reliable since the author reported that TIPS-7B (41.0) performs worse than TIPS-4B (48.0)
> > >
> > > The 7B model is Qwen 2.5 while the 4B model is Qwen 3, as specified in our table. Qwen3-4B was extensively trained with RL for reasoning and tool use, while Qwen 2.5 was not. **As a base model, it outperforms Qwen2.5-7B on essentially all benchmarks** [1] [2]. Given this, the observed ordering is unsurprising: **Qwen3-4B exceeds Qwen2.5-7B at every setting—baseline (28.4 vs. 25.2), PPO (45.1 vs. 37.3), and TIPS (48.4 vs. 41.7).** For our argument, what ultimately matters is **that TIPS exceeds PPO on every model**, which our results confirm (see table in A8).
> > >
> > > > the marginal performance improvements along with high computational costs make the practical value of the method limited
> > > >
> > >
> > > We do not agree that the computational cost is high, nor that the improvements are marginal. We’ve conducted a detailed overhead analysis and found a ~12% FLOPs overhead of TIPS compared to PPO. For context, GRPO—a standard baseline—has a 350% FLOPs overhead. In LLM RL, generation is *by far* the dominating factor in runtime, as it does 1 forward pass *per generated token.* TIPS adds only 1 forward pass *per turn*, which consist of hundreds to thousands of tokens. Given this small cost, we see consistent and significant improvements in stability and performance across model sizes and families, as discussed in A8. Note that reward collapses in RL are usually irrecoverable; any technique that prevents this—such as TIPS—is of **immense interest to practitioners.**
> > >
> > > > the reported results in Table 1 are inconsistent with those in the cited Search-R1 paper. Although the author provided some expatiation, I don’t think it is convincing since the results of Search-R1 is reproducible.
> > > >
> > >
> > > As discussed in A7, the **Search-R1 results are not consistently reproducible**. The **Search-R1 authors themselves acknowledge instability issues** in the paper [3] and on GitHub [4] [5]. Our numbers fall within the performance range reported by independent reproductions of Search-R1 in the field [6] [7] [8]. **This only increases our confidence in the quality of our results.**
> > >
> > > We collect results from reproductions in the table for reference, all other papers are using same FLOPs if not specified:
> > >
> > > | **Search-R1 results as reproduced in the following papers** | **Qwen2.5-7B-Instruct (EM)** | **Qwen2.5-3B-Instruct (EM)** |
> > > | --- | --- | --- |
> > > | **Search-R1 [3]** | 38.5 | 32.5 |
> > > | **RAG-R1 [6]** | 36.8 | - |
> > > | **Scent of Knowledge [7]** | - | 31.8 |
> > > | **ZeroSearch [8]** | 38.1 | 28.1 |
> > > | **Ours (halved batch size)** | 37.3 | 30.1 |
> > >
> > > We would like to highlight that we achieve better performance than Search R1’s reported results using TIPS with only ~56% the FLOPs, as they use double our batch size with the same number of training steps.
> > >
> > > | Model | PPO (Search-R1, TFLOPs/step) | PPO (Search-R1, EM) | TIPS (TFLOPs/step) | TIPS (EM) |
> > > | --- | --- | --- | --- | --- |
> > > | Qwen2.5-3B-Instruct | 128k | 32.5 | 72k | 33.6 |
> > > | Qwen2.5-7B-Instruct | 272k | 38.5 | 152k | 41.7 |
> > >
> > > ---
> > >
> > > [1] Yang et al. **Qwen3 Technical Report.** arXiv:2505.09388, 2025.
> > >
> > > [2] Qwen Team et al. **Qwen2.5 Technical Report.** arXiv:2412.15115, 2024.
> > >
> > > [3] Jin et al. **Search-R1.** arXiv:2503.09516, 2025.
> > >
> > > [4] scris. **Search-R1 issue #88.** GitHub, 2025.
> > >
> > > [5] thakur-nandan. **Search-R1 issue #142.** GitHub, 2025.
> > >
> > > [6] Tan et al. **RAG-R1.** arXiv:2507.02962, 2025.
> > >
> > > [7] Qian & Liu. **Scent of Knowledge.** NeurIPS 2025.
> > >
> > > [8] Sun et al. **ZeroSearch.** arXiv:2505.04588, 2025.

---

> > > > ### Comment · Reviewer_ajMR · 2025-11-25
> > > >
> > > > Thank you for this further response. I'd like to retain my score since my key concerns about computational efficiency, robustness, and generalizability are not fundamentally addressed
> > > >
> > > > Efficiency issue: Your rebuttal justifies the computational overhead of TIPS mostly by comparing it with PPO and GRPO. This is an insufficient frame of reference. The critical question is not whether TIPS is more efficient than notoriously expensive baselines like PPO and GRPO, but whether its performance gains justify its own added cost over the standard paradigm. Additionally, a consistent ~12% FLOPs overhead for what are, in several cases, marginal absolute improvements, e.g., +3.5 EM on the 3B model, does not present a compelling cost-benefit ratio. The efficiency argument remains unconvincing.
> > > >
> > > > Robustness and generalizability issues: My biggest concern is with the core mechanism of your method. The turn-level reward is obtained by measuring the change in the likelihood of a pre-defined set of acceptable answers given by a teacher model. In this respect, the design is inherently fragile. TIPS would suffer from a partly incomplete, imperfect, or overly narrow set of answers. In open-domain QA, where paraphrasing answers is common, reliance on a fixed set A introduces a major point of failure and potential biases not validated. The reliability of your whole reward-shaping framework is thus questionable.
> > > >
> > > > For these reasons, I believe that the paper needs major revision to address these foundational issues before it can be considered for acceptance.

---

> ### Author Response · Authors · 2025-11-26
> **Thank you!**
>
> Thanks for your response! We appreciate you taking the time to read our responses and raise further concerns.
>
> ---
>
> > Your rebuttal justifies the computational overhead of TIPS mostly by comparing it with PPO and GRPO. This is an insufficient frame of reference. The critical question is not whether TIPS is more efficient than notoriously expensive baselines like PPO and GRPO, but whether its performance gains justify its own added cost over the standard paradigm.
> >
>
> The standard paradigm in RLVR for LLMs is GRPO and PPO, as shown in [1-12]. They are indeed expensive, but that is due to the compute intensive nature of training LLMs. We stand by our use of these algorithms as our baselines.
>
> > Additionally, a consistent ~12% FLOPs overhead for what are, in several cases, marginal absolute improvements, e.g., +3.5 EM on the 3B model, does not present a compelling cost-benefit ratio.
> >
>
> Out of our experiments across several model sizes and families, Qwen2.5-3B had the smallest improvement relative to PPO. Yet, it still outperformed the PPO baseline. **We ask the reviewer to consider our results across *all experiments* instead of focusing on the least performant one when judging the impact of TIPS.**
> Another core contribution of TIPS is **improved training stability.** In our runs, vanilla PPO frequently undergoes irrecoverable reward collapse, after which **additional compute no longer improves performance**, whereas TIPS maintains stable learning and continues to convert extra FLOPs into higher accuracy.
>
> We maintain that paying a ≈12% increase in training FLOPs for consistent gains (e.g., +7–14% on the Qwen family and +29–34% on LLaMA3.1-8B) and avoiding irrecoverable reward collapses is a reasonable trade-off.
>
> > TIPS would suffer from a partly incomplete, imperfect, or overly narrow set of answers. In open-domain QA, where paraphrasing answers is common, reliance on a fixed set A introduces a major point of failure and potential biases not validated.
> >
>
> All other RLVR methods on verifiable multi-turn QA tasks rely on the finite gold answer set per question [1-9]; **TIPS is not more sensitive to answer set incompleteness than the baselines, as discussed in A3.** Any incompleteness or narrowness in the answer sets therefore affects the RLVR framework as a whole, not TIPS specifically.
>
> We would like to remind the reviewer of the purpose and scope of our paper: we are focused on introducing a novel, dense credit assignment method for multi-turn RLVR and validating it in a controlled environment. We are not trying to build a general search agent, rebuild standard datasets, or use non-verifiable rewards.
>
> ---
>
> [1] Jin et al. **Search-R1.** arXiv:2503.09516, 2025.
>
> [2] Chen et al. **ReSearch.** arXiv:2503.19470, 2025.
>
> [3] Gao et al. **Beyond Ten Turns.** arXiv:2508.07976, 2025.
>
> [4] Guan et al. **DeepRAG.** arXiv:2502.01142, 2025.
>
> [5] Li et al. **Search-o1.** arXiv:2501.05366, EMNLP 2025.
>
> [6] Song et al. **R1-Searcher.** arXiv:2503.05592, 2025.
>
> [7] Tan et al. **RAG-R1.** arXiv:2507.02962, 2025.
>
> [8] Qian & Liu. **Scent of Knowledge.** NeurIPS 2025.
>
> [9] Sun et al. **ZeroSearch.** arXiv:2505.04588, 2025.
>
> [10] Ouyang et al. **Training language models to follow instructions with human feedback.** arXiv:2203.02155, NeurIPS 2022.
>
> [11] Shao et al. **DeepSeekMath: Pushing the limits of mathematical reasoning in open language models.** arXiv:2402.03300, 2024.
>
> [12] Guo et al. **DeepSeek-R1: Incentivizing reasoning capability in LLMs via reinforcement learning.** arXiv:2501.12948, 2025.

---

### Official Review · Reviewer_VPap · 2025-10-31

**Soundness:** 3
**Presentation:** 3
**Contribution:** 2
**Rating:** 4
**Confidence:** 2

**Summary:**

This paper introduces TIPS (Turn-Level Information-Potential Reward Shaping), a novel method to address the sparse reward and credit assignment problem in training multi-turn, search-augmented Large Language Models (LLMs). The core idea is to provide dense, turn-level rewards by measuring the increase in the likelihood of a correct answer, as evaluated by a "teacher" model, after each reasoning-and-retrieval turn. This information-gain signal is framed as a Potential-Based Reward Shape (PBRS), which theoretically preserves policy invariance. The authors integrate TIPS into the PPO framework and evaluate it on seven open-domain QA benchmarks. Results show that TIPS consistently outperforms strong baselines like PPO and GRPO, achieving significant improvements in Exact Match and F1 scores (e.g., +11.8% EM and +13.6% F1 for a 7B model) while demonstrating superior training stability and more effective credit assignment.

**Strengths:**

Originality: The core idea of using a teacher model's likelihood of the correct answer to compute information-gain rewards is highly original. It provides a principled and automated way to generate dense supervision, distinct from heuristic rules or learned reward models.

Quality: The work is of very high quality. The combination of a solid theoretical grounding (PBRS) with extensive and carefully designed empirical validation is commendable. The ablations and analysis sections are particularly thorough.

Clarity: The paper is a model of clarity. The writing is precise, the concepts are well-motivated, and the flow from problem statement to solution to validation is logical and easy to follow.

Significance: This work addresses a fundamental bottleneck in RLHF/RLVR for complex LLM tasks. The demonstrated improvements in performance and, crucially, in training stability, represent a significant advance. The method has the potential to become a standard component in the toolkit for training reasoning agents.

**Weaknesses:**

The weaknesses are minor and do not detract from the overall excellent contribution.

1. Computational Overhead: While not explicitly quantified, using a teacher model (especially a 7B model) to compute log-likelihoods for every turn during training introduces non-trivial computational overhead compared to outcome-only rewards. A brief discussion of this cost (e.g., estimated % increase in training time or FLOPs) would be helpful for practitioners.

2. Teacher-Student Capacity: The method assumes the teacher model is sufficiently powerful to make reliable judgments about information gain. The paper uses the agent's own initial checkpoint as the teacher. It is unclear how the method would perform if the agent significantly diverges from the teacher's knowledge distribution, or if a less capable teacher were used. An ablation with teachers of different capacities could further strengthen the work.

3. Comparison to PRMs: The introduction contrasts TIPS with Process Reward Models (PRMs) by citing their need for "high-quality supervisory labels." However, recent work has explored training PRMs on synthetic or LLM-generated step-level feedback. A more direct comparison or discussion on how TIPS's "information gain" reward compares in quality and efficiency to a learned PRM would be insightful.

It is worth noting that I am not an expert in this field, and I will adjust my score based on the scores of other reviewers.

**Questions:**

1. Computational Cost: Could the authors provide an estimate of the computational overhead introduced by the teacher model's forward passes for calculating $I_k$? How does the wall-clock time or FLOPs for a TIPS training run compare to a standard PPO run?

2. Teacher Model Selection: The paper uses the agent's initial checkpoint as the teacher. Did you experiment with using a larger, more powerful frozen model as the teacher? If so, what were the results? Conversely, what happens if the teacher is significantly weaker than the agent being trained?

3. Theoretical Guarantees and Teacher Refresh: The policy invariance proof holds for a fixed potential function $\Phi$. Your refresh@200 strategy, while empirically beneficial, technically breaks this strict invariance between refresh windows. Can you comment on the theoretical implications of this? Did you observe any instability or "jumps" in the return values immediately after a refresh, and how does this relate to the "constant shift" mentioned in the text?

4. Failure Modes: The advantage distribution (Fig. 5) for TIPS is much cleaner than PPO's. However, are there specific types of tasks or reasoning steps where the information-gain reward might provide misleading guidance (e.g., when new evidence is subtly distracting but doesn't immediately lower the teacher's likelihood)? Can you provide an example of a failure case for TIPS?

---

> ### Author Response · Authors · 2025-11-23
> **Thank you!**
>
> We thank the reviewer for their valuable feedback. We address your comments in the following.
>
> ---
>
> **W1/Q1**: Computational Cost: Could the authors provide an estimate of the computational overhead introduced by the teacher model's forward passes for calculating? How does the wall-clock time or FLOPs for a TIPS training run compare to a standard PPO run?
>
> **A1:** We agree that quantifying the extra cost is important, and we have **added an Overhead Analysis section (App. H) with both FLOPs and wall-clock measurements.**
>
> Using the FLOPs accounting from the VeRL codebase (including the extra teacher forward passes with KV-prefix caching), we find that TIPS increases the training FLOPs by only about **11–12%** over vanilla PPO across all tested backbones:
>
> | Model | EM (Δ vs PPO) | F1 (Δ vs PPO) | FLOPs overhead |
> | --- | --- | --- | --- |
> | Qwen2.5-3B-Instruct | 33.6 (+11.4%) | 41.1 (+10.2%) | 11.76% |
> | Qwen3-4B-Instruct | 48.4 (+7.3%) | 57.1 (+6.1%) | 11.85% |
> | Qwen2.5-7B-Instruct | 41.7 (+11.9%) | 51.2 (+13.7%) | 11.81% |
> | Qwen2.5-14B-Instruct | 45.4 (+12.7%) | 53.1 (+10.6%) | 11.81% |
> | LLaMA3.1-8B | 40.3 (+34.0%) | 49.0 (+29.3%) | 11.66% |
>
> For wall-clock time, rollout generation dominates, so per-step runtime is largely driven by the policy’s average response length. To isolate the fixed overhead from teacher scoring, we measure the per-step time with reward computation on vs. off within the same TIPS run:
>
> | Model | Without reward (s/step) | With reward (s/step) | Overhead (%) |
> | --- | --- | --- | --- |
> | Qwen2.5-3B | 30.56 | 35.95 | 17.64 |
> | Qwen2.5-7B | 93.00 | 108.22 | 16.37 |
>
> Thus, teacher scoring adds roughly **11–12% training FLOPs** and **16–17% per-step wall-clock time**, with **no additional inference cost** (the teacher is not used at inference).
>
> Given that TIPS yields **7–14% relative EM/F1 gains** on the Qwen family and **29–34% relative EM/F1 gains** on LLaMA3.1-8B, together with substantially improved training stability and reduced collapse rates, we believe this is a favorable cost–benefit trade-off.
>
> ---
>
> **W2/Q2**: Teacher–student capacity and teacher selection: The method assumes the teacher is sufficiently powerful and uses the agent’s own initial checkpoint as teacher. How does TIPS behave if the policy diverges from the teacher, or if the teacher is weaker/stronger? An ablation over teacher capacities would help.
>
> **A2.** TIPS does **not** require the teacher to be strictly stronger than the policy, nor to make “objectively correct” judgments. The key requirement is that the teacher’s distribution is **well aligned** with the policy’s distribution so that the teacher’s log-likelihoods over valid answers can act as a stable potential function.
>
> In all main experiments we therefore choose the teacher to be a **frozen copy of the policy**, refreshed periodically. This guarantees availability at any scale (no extra model to train) and keeps the teacher and policy close in behavior.
>
> To directly address the reviewer’s question, we also ran an ablation where we fix the policy and vary the teacher:
>
> - **Frozen policy (self-teacher)** — our default;
> - **Qwen3-4B-TIPS** — a stronger, TIPS-trained model;
> - **LLaMA3.1-8B** — a cross-family model.
>
> The EM results on the multi-turn QA suite are:
>
> | Policy / Teacher | Frozen policy | Qwen3-4B-TIPS | LLaMA3.1-8B |
> | --- | --- | --- | --- |
> | Qwen2.5-7B | 41.7 (+11.9%) | 30 (−19.5%) | 29 (−22.2%) |
> | Qwen3-4B | 48.4 (+7.3%) | 45.88 (+1.7%) | 43 (−4.7%) |
>
> Using a **frozen copy of the policy** as teacher consistently performs best. Even a stronger TIPS-trained Qwen3-4B teacher underperforms the self-teacher, and a cross-family LLaMA teacher degrades performance further. This supports our design choice: for TIPS, **behavioral similarity between teacher and policy is more important than raw teacher capacity**.
>
> We have added this ablation and a brief discussion of  **Teacher selection** to Section 4.3 of the revised manuscript. ([screenshot](https://imgur.com/a/6rclzlc)).
>
> **(1/3)**

---

> ### Author Response · Authors · 2025-11-23
>
> **W3**: Comparison to PRMs: The introduction contrasts TIPS with Process Reward Models (PRMs) by citing their need for "high-quality supervisory labels." However, recent work has explored training PRMs on synthetic or LLM-generated step-level feedback. A more direct comparison or discussion on how TIPS's "information gain" reward compares in quality and efficiency to a learned PRM would be insightful.
>
> **Response:** Thank you for pointing this out. Our intention in the introduction was not to claim that PRMs are infeasible, but to highlight a complementary design point: TIPS aims to provide a very low-cost, plug-in reward shaping signal, whereas PRMs require training an additional model from step-level supervision (human, synthetic, or LLM-generated).
>
> Concretely, a fair experimental comparison to PRMs in our setting would require (i) generating or collecting step-level labels for multi-turn web-search traces, (ii) training a separate process reward model on this data, and (iii) running RL with an extra PRM forward pass at each step, on top of the existing verifier calls. To the best of our knowledge, there is currently no off-the-shelf PRM or any relevant dataset for our multi-turn QA / RLVR environment, and building such a pipeline would be a substantial project on its own, orthogonal to the core contribution of this paper.
>
> By contrast, TIPS reuses a frozen copy of the policy as the “teacher” and leverages only existing outcome rewards; it introduces no additional training phase and adds a **relatively small runtime overhead (≈11–12% FLOPs, see Appendix. H).** Our goal is to show that this  **simple information-gain–based shaping already yields finer credit assignment** and more stable RL over PPO/GRPO.
>
> **(2/3)**

---

> ### Author Response · Authors · 2025-11-23
>
> **Q3**: Theoretical Guarantees and Teacher Refresh: The policy invariance proof holds for a fixed potential function. Your refresh@200 strategy, while empirically beneficial, technically breaks this strict invariance between refresh windows. Can you comment on the theoretical implications of this? Did you observe any instability or "jumps" in the return values immediately after a refresh, and how does this relate to the "constant shift" mentioned in the text?
>
> **A3**: Our formal proof indeed assumes a fixed potential $\Phi$, and we now clarify how this interacts with teacher refresh.
>
> From a theoretical perspective:
> For any **given rollout**, the teacher (and thus  $\Phi$) is **frozen**, and no gradients flow through $\Phi$. For that trajectory, the shaped return for token (t) in turn (k) can be written as:
>
> $$
>     G_t^{(R+I)} = G_t^{(R)} - \alpha\Phi(S_{k-1}),
> $$
>
> so the shaping term adds a **trajectory-dependent constant** that does *not* change the relative advantages within that rollout. PBRS invariance therefore holds **per trajectory**.
>
> When we refresh, we copy the current policy into the teacher and use the new $\Phi$ only for **future** trajectories. This behaves like periodically updating a **baseline between batches**: it shifts the overall scale of returns across batches, but does not alter the within-trajectory ordering that drives policy improvement.
>
> We refresh because TIPS relies on **alignment between teacher and policy distributions**. If the teacher is never updated, it becomes increasingly stale as the policy improves, and changes in $\Phi$ no longer reliably track what is useful for the current policy. If we refresh too frequently, $\Phi$ changes very rapidly and the baseline moves often, which mainly affects the **critic’s stability** rather than the underlying PBRS property.
>
> To study this trade-off, we ablate the refresh interval (N $\in$ {1, 100, 200, 250, 500}) for Qwen2.5-7B and Qwen3-4B in the **Refresh interval** of Sec. 4.3:
>
> - Very rare refreshes ((N{=}500), effectively a fixed teacher) yield noticeably lower EM.
> - There is a broad optimum around (N $\approx$ 200): Qwen2.5-7B reaches (41.7) EM and Qwen3-4B reaches (48.4) EM.
> - Across the full range of moderate intervals ((100)–(500)), all TIPS runs still outperform PPO, indicating that TIPS does *not* rely on a finely tuned refresh rate.
> [Screenshots of Fixed v.s. Refresh interval ablation](https://imgur.com/a/Z7zs3bo)
>
> Empirically, we do observe a **small change in mean reward scale** ([W&B Plot](https://imgur.com/a/N4YkRkl)) right after refresh caused by the change of information reward scale, but **the critic loss ([W&B Plot](https://imgur.com/a/KQHm5c5)) and gradient norms ([W&B Plot](https://imgur.com/a/g9u3vCS)) remain stable and quickly re-equilibrate.** We only observe a **slight shift in the critic values** ([W&B Plot](https://imgur.com/a/lF38GyO)); critic training remains stable. And we do not see refresh-induced divergence or collapses.
>
> We have added this discussion, the refresh-interval ablation, and the corresponding plots in the **Refresh interval** of Sec. 4.3 in the revised manuscript.
>
>
> -----
>
> **Q4**: Failure Modes: The advantage distribution (Fig. 5) for TIPS is much cleaner than PPO's. However, are there specific types of tasks or reasoning steps where the information-gain reward might provide misleading guidance (e.g., when new evidence is subtly distracting but doesn't immediately lower the teacher's likelihood)? Can you provide an example of a failure case for TIPS?
>
> **A4**: We agree that characterizing failure modes for TIPS is important. The most common failure case we observe is when the base model already knows the answer to the question from pre-training, and a tool use only distracts the model. In this scenario, we often find negative rewards assigned to perfectly good tool calls. However, we found that the magnitude of these rewards are usually small compared to the outcome reward, and don’t appear to actively harm training. We have provided an example of this in Appendix G.2, Figure 12 ([screenshot](https://imgur.com/a/6M8xfUz)).
>
> -----
>
> We believe that our response and additional experiment results address your concerns. However, please do not hesitate to let us know if you have any additional comments.
>
> **(3/3)**

---

> ### Author Response · Authors · 2025-11-26
> **Thank you!**
>
> Dear reviewers,
>
> Thank you again for the time and effort you have put into reviewing our submission and engaging in the discussion so far. **As the rebuttal and discussion period will conclude in about one week**, we would like to make sure that we have properly addressed your concerns before the deadline.
>
> If there are any remaining questions or points that you feel are not yet fully clarified, please let us know — we are very happy to provide further details or additional clarification while the discussion is still open.
>
> Best regards,
>
> Authors of *TIPS*

---

### Official Review · Reviewer_vPWX · 2025-10-31

**Soundness:** 3
**Presentation:** 3
**Contribution:** 3
**Rating:** 6
**Confidence:** 3

**Summary:**

This paper presents TIPS (Turn-level Information-Potential Shaping), a reinforcement learning framework for optimizing search-augmented large language models in open-domain question answering. The authors identify that existing reinforcement learning methods such as PPO and GRPO depend heavily on sparse, outcome-only rewards that are only provided at the end of an episode. This design makes it difficult to assign credit accurately across multiple reasoning and retrieval steps, leading to unstable learning dynamics.

TIPS addresses this limitation by introducing a turn-level dense reward mechanism. Instead of waiting for a final correctness signal, the framework measures how much each reasoning-and-retrieval step increases the teacher model’s confidence in the correct answer. This reward, based on the change in the teacher’s confidence, follows the Potential-Based Reward Shaping (PBRS) principle, which guarantees that the underlying task objective remains unchanged while providing richer learning feedback. The system therefore offers fine-grained credit assignment across reasoning turns without compromising theoretical soundness.

The authors evaluate TIPS on seven open-domain QA benchmarks (including NQ, TriviaQA, 2Wiki, MuSiQue, and HotpotQA) using Qwen-2.5 models of 3B and 7B parameters. Across tasks, TIPS consistently surpasses PPO and GRPO in both accuracy and stability, with particularly strong improvements on multi-hop reasoning datasets.

**Strengths:**

- Built on potential-based reward shaping, TIPS ensures that policy invariance is maintained while providing denser feedback signals, addressing a fundamental limitation in sparse-reward reinforcement learning for language models.
- The paper presents a well-structured pipeline—multi-turn reasoning, retrieval, teacher evaluation, and potential-based shaping—supported by consistent mathematical logic and implementation clarity.
- Evaluations across seven QA benchmarks and two model scales demonstrate reproducible and significant improvements in EM/F1 metrics, convergence speed, and training stability.
- The inclusion of advantage-distribution analysis and ablation studies on shaping coefficients and teacher refresh strategies provides concrete evidence of behavioral differences and system robustness.
- The framework is directly applicable to retrieval-augmented and reasoning-intensive LLMs, making it relevant for real-world agentic settings that require iterative tool use.

**Weaknesses:**

- The reward signal is fully determined by the teacher model’s likelihood estimates. If the teacher is miscalibrated or biased, the shaping signal may misrepresent information gain. No calibration analysis or correction mechanism is discussed.
- All experiments use the same teacher model (Qwen-2.5), differing only in whether it is fixed or periodically refreshed. The paper does not evaluate how the reward behaves with different teachers, leaving the robustness of TIPS to teacher variation untested.
- The need to query the teacher model for every reasoning turn increases training cost compared to standard PPO/GRPO. The paper does not quantify the additional compute or discuss trade-offs between stability and efficiency.
- All results are reported using EM/F1 metrics. There is no qualitative or human study to determine whether TIPS improves reasoning coherence, interpretability, or factual correctness in practice.

**Questions:**

see weakness

---

> ### Author Response · Authors · 2025-11-23
> **Thank you!**
>
> We thank the reviewer for their valuable feedback. We address your comments in the following.
>
> ---
>
> **Q1**: The reward signal is fully determined by the teacher model’s likelihood estimates. If the teacher is miscalibrated or biased, the shaping signal may misrepresent information gain. No calibration analysis or correction mechanism is discussed.
>
> **A1**: This is a very important point that must be tackled when implementing our potential function for PBRS. On one hand, we want to keep the potential signal stationary for training stability. On the other hand, we want to ensure that the potential function doesn’t become stale, or miscalibrated in your words. Our main method to prevent staleness is refreshing the teacher model from the latest policy checkpoint every n iterations. Our experiments in Section 4.3 show that your concern is valid; if we do not refresh the teacher, the biased potential causes the reward curve to collapse. However, when we do refresh the teacher, reward continues to climb. We have updated the manuscript to discuss the relationship between teacher calibration and refreshing the teacher ([screenshot](https://imgur.com/a/zhkOgyn)).
>
> ---
>
> **Q2**: All experiments use the same teacher model (Qwen-2.5), differing only in whether it is fixed or periodically refreshed. The paper does not evaluate how the reward behaves with different teachers, leaving the robustness of TIPS to teacher variation untested.
>
> **A2**: Thank you for pointing this out. TIPS was designed with the intention of using the same model as the policy for the teacher. The teacher is simply a frozen checkpoint of the policy model being trained, refreshed at a fixed interval. Because the teacher and policy distributions are similar, the probability the teacher assigns to generating an answer may act as a **potential**, with which we can assign intermediate credit to the policy. However, we agree that exploring the use of other models as teachers is worthwhile. Our ablations yield the following results:
>
> | Policy/Teacher | Frozen policy | Qwen3-4B-TIPS | Llama3.1-8B |
> | --- | --- | --- | --- |
> | Qwen2.5-7B  | 41.7  (+11.9%) | 30   (-19.5%) | 29  (-22.2%) |
> | Qwen3-4B   | 48.4  (+7.3%) | 45.88 (+1.7%) | 43  (-4.7%) |
>
> We see that using a frozen policy consistently performs best. Notably, it outperforms using a *stronger model* as a teacher. This supports our hypothesis that the similarity between the policy and teacher matters much more than the teacher’s strength.
>
> We have added these results and a brief discussion of teacher selection to Section 4.3 ([screenshot](https://imgur.com/a/6rclzlc)).
>
> --------
>
> **Q3**: The need to query the teacher model for every reasoning turn increases training cost compared to standard PPO/GRPO. The paper does not quantify the additional compute or discuss trade-offs between stability and efficiency.
>
> **A3**: We agree that this is a crucial point that we neglected to analyze. We have updated the manuscript with a report of the computational overhead of TIPS over the baselines, and a thorough analysis in Appendix H.
>
> When we compute the increase in FLOPs attributed to the teacher forward passes, we find a modest overhead of ~12% compared to PPO. To put this in context, GRPO requires 3.5x more FLOPs than PPO.
>
> Comparing runtime between TIPS and baselines is more challenging. Since rollout generation is the slowest phase of online LLM RL, the per-step runtime is mostly determined by the policy’s average response length. To get an estimate, however, we can compute (step time with reward)/(step time without reward). This yields an overhead of 16-17%.
>
> | Model | Without Reward (s/step) | With Reward (s/step) | Overhead (%) |
> | --- | --- | --- | --- |
> | Qwen2.5-3B | 30.56 | 35.95 | 17.64 |
> | Qwen2.5-7B | 93.00 | 108.22 | 16.37 |
>
> Although the overhead is real, it is quite modest, and we believe the improvement in performance and stability is well worth this cost.
>
> **(1/2)**

---

> ### Author Response · Authors · 2025-11-23
>
> **Q4**: All results are reported using EM/F1 metrics. There is no qualitative or human study to determine whether TIPS improves reasoning coherence, interpretability, or factual correctness in practice.
>
> **A4**: In the search engine QA task we use for evaluation, there is not much room to evaluate reasoning coherence, interpretability, or factual correctness. The queries in our dataset have specific, unambiguous answers that are usually 1-3 words; i.e. they are **not open ended**. We have updated the Problem Formulation section to make this clear ([screenshot](https://imgur.com/a/OoTD1S0)). Although this task isn’t representative of fully general, real world use, it is an excellent target for RL with verified rewards. We therefore stand by the use of EM and F1 as the best evaluation metrics given our dataset. Yet, we recognize the value of qualitative case studies. Since the primary contribution of TIPS is improving credit assignment, we point the reviewer to the critic network heatmaps we included in the Appendix G. We did extensively comb through rollout samples from TIPS compared to baselines, and did not find any notable properties that were not reflected in the reported numbers: TIPS had more successful tool calls and less jumbled output caused by training instability.
>
> -----
>
> We believe that our response and additional experiment results address your concerns. However, please do not hesitate to let us know if you have any additional comments.
>
> **(2/2)**

---

> ### Author Response · Authors · 2025-11-26
> **Thank you!**
>
> Dear reviewers,
>
> Thank you again for the time and effort you have put into reviewing our submission and engaging in the discussion so far. **As the rebuttal and discussion period will conclude in about one week**, we would like to make sure that we have properly addressed your concerns before the deadline.
>
> If there are any remaining questions or points that you feel are not yet fully clarified, please let us know — we are very happy to provide further details or additional clarification while the discussion is still open.
>
> Best regards,
>
> Authors of *TIPS*

---

### Official Review · Reviewer_EXdd · 2025-11-01

**Soundness:** 3
**Presentation:** 2
**Contribution:** 2
**Rating:** 4
**Confidence:** 3

**Summary:**

The paper introduces Turn-Level Information Potential Reward Shaping (TIPS), a simple framework that assigns dense, turn-level rewards to each reasoning + tool-call segment based on the increased likelihood of the correct answer under a teacher model.

**Strengths:**

1. The motivation of this paper is sound: it aims to introduce denser reward signals to improve RL training.
2. The experimental results show notable improvements.

**Weaknesses:**

1. The writing quality needs improvement: Section 3 is hard to follow, and I could not find any mention of what exact model is as the teacher model (if I did not miss anything).

2. The experimental setup seems outdated. Why not evaluate on GAIA or BrowseComp for search LLM? Likewise, why stick with the Qwen2.5 series, which is barely capable of search, instead of building upon the latest Qwen3 or other up-to-date models?

3. The proposed method is heavily tied to a teacher model. A fairer baseline would be direct distillation from that same teacher.

**Questions:**

See above.

---

> ### Author Response · Authors · 2025-11-23
> **Thank you!**
>
> We thank the reviewer for their valuable feedback. We address your comments in the following.
>
> -------
>
>
> **Q1:** The writing quality needs improvement: Section 3 is hard to follow, and I could not find any mention of what exact model is as the teacher model (if I did not miss anything).
>
> **A1:** We apologize for the lack of clarity. The “teacher” model is simply a frozen checkpoint of the policy that is refreshed at a fixed interval from the policy as is it trained. It is initialized from the base policy model. We chose this so that the probability assigned to generating an answer by the teacher shares a similar distribution to the policy, and therefore can be used as an effective potential function. The teacher is frozen to keep the potential stable. Using the policy model as a teacher keeps our method scalable, as it does not require a stronger external model while providing dynamically updated guidance. We have updated Section 3 in the manuscript to significantly improve clarity, moving distracting formal definitions to the appendix and adding clear, high level descriptions of our method. We hope these changes resolve your concerns.
>
> ---
>
> **Q2**: The experimental setup seems outdated. Why not evaluate on GAIA or BrowseComp for search LLM? Likewise, why stick with the Qwen2.5 series, which is barely capable of search, instead of building upon the latest Qwen3 or other up-to-date models?
>
> **A2**: Thank you for raising this concern. We agree that evaluating on modern search benchmarks and more recent model families is important for assessing the practical relevance of TIPS.
>
> Our original experiments followed the Search-R1-style setup on multi-turn Wikipedia QA with Qwen2.5 models because this environment provides a controlled, fully verifiable RLVR setting, and Qwen2.5-Instruct has not been heavily tuned for tool use, which allows us to isolate the effect of reward shaping on credit assignment rather than confounding it with a sophisticated search-agent design.
>
> To address your suggestion, we add two new groups of experiments in the revision:
>
> 1. **BrowseComp-Plus.**
>
>     BrowseComp-Plus is a controlled variant of BrowseComp that replaces the live web with a fixed curated corpus and a shared BM25 retriever, enabling fair and reproducible comparison of deep-research agents. We follow its protocol by plugging our PPO/GRPO/TIPS-trained models in as the LLM component, keeping the corpus and retriever fixed, and reporting the official exact-match accuracy. As shown in Fig. 7 and Table 7, TIPS consistently improves over PPO/GRPO; in particular, Qwen3-4B-TIPS reaches 9.40% accuracy, substantially outperforming the reported Search-R1-32B agent (3.86%) under the same BM25 setting despite using a much smaller model.
>
> 2. **Model families and scales.**
>
>     We further train TIPS on Qwen3-4B-Instruct, Qwen2.5-14B-Instruct, and LLaMA-3.1-8B. Across our multi-turn QA suite, TIPS improves EM/F1 over PPO for all these backbones, showing that it is not tied to the Qwen2.5 family or a specific model size and that it yields meaningful gains on a modern deep-research benchmark.
>
>
> Regarding GAIA, it targets multi-tool, multimodal assistants with live web access and code execution, whereas we study a single-tool, text-only RLVR setting with a fixed Wikipedia search API to isolate the effect of reward shaping. Building a full GAIA-style agent is beyond the scope of this work and we leave it for future work; our BrowseComp-Plus and cross-family experiments already suggest that TIPS transfers to modern web-search scenarios.
>
> [**Screenshot of Figure 7**](https://imgur.com/a/x8AqdNx)
>
> **BrowseComp Plus Results**
>
> | Model | PPO Accuracy (%) | GRPO Accuracy (%) | TIPS Accuracy (%) |
> | --- | --- | --- | --- |
> | Qwen2.5-3B-Instruct | 1.2 | 1.2 | 1.57 |
> | Qwen2.5-7B-Instruct | 1.24 | 3.61 | 4.1 |
> | Qwen3-4B-Instruct | 6.75 | 2.65 | 9.4 |
> | Search-r1 32B | 4.11 | — | — |
>
> **Main results**
> | Model | EM | F1 |
> | --- | --- | --- |
> | Qwen2.5-3B-Instruct | 33.6  (+11.4%) | 41.1  (+10.2%) |
> | Qwen3-4B-Instruct | 48.4  (+7.3%) | 57.1  (+6.1%) |
> | Qwen2.5-7B-Instruct | 41.7  (+11.9%) | 51.2  (+13.7%) |
> | Qwen2.5-14B-Instruct | 45.4  (+12.7%) | 53.1  (+10.6%) |
> | LLaMA3.1-8B | 40.3  (+34.0%) | 49.0  (+29.3%) |
>
>  **(1/2)**

---

> ### Author Response · Authors · 2025-11-23
>
> **Q3**: The proposed method is heavily tied to a teacher model. A fairer baseline would be direct distillation from that same teacher.
>
> **A3**: In TIPS, the teacher is simply a frozen checkpoint of the policy model being trained, refreshed at a fixed interval. Because the teacher and policy distributions are similar, the probability the teacher assigns to generating an answer may act as a **potential**, with which we can assign intermediate credit to the policy. We believe your question assumes our teacher is a stronger model (as the name may imply), but this is not the case in TIPS. We apologize for the lack of clarity on this matter, and have updated the manuscript to make these details more apparent.
>
> -----
>
> We believe that our response and additional experiment results address your concerns. However, please do not hesitate to let us know if you have any additional comments.
>
> **(2/2)**

---

> ### Author Response · Authors · 2025-11-26
> **Thank you!**
>
> Dear reviewers,
>
> Thank you again for the time and effort you have put into reviewing our submission and engaging in the discussion so far. **As the rebuttal and discussion period will conclude in about one week**, we would like to make sure that we have properly addressed your concerns before the deadline.
>
> If there are any remaining questions or points that you feel are not yet fully clarified, please let us know — we are very happy to provide further details or additional clarification while the discussion is still open.
>
> Best regards,
>
> Authors of *TIPS*

---

### Author Response · Authors · 2025-11-23
**General comment**

**We thank all reviewers for their thoughtful comments and have revised our manuscript based on your feedback. We have also responded to individual comments in detail.**

**Summary of revisions (highlights in green in the new version):**

* ([EXdd](https://openreview.net/forum?id=eBMOr6a84z&noteId=khNAqnKkLY)) Rewrote Sec. 3 for clarity, moving most formal MDP/PBRS details to the appendix and replacing them with a higher-level, step-by-step description of TIPS and its intuition.
* ([EXdd](https://openreview.net/forum?id=eBMOr6a84z&noteId=khNAqnKkLY), [vPWX](https://openreview.net/forum?id=eBMOr6a84z&noteId=sw5H21H2ll), [VPap](https://openreview.net/forum?id=eBMOr6a84z&noteId=I4l3TcOFDQ), [ajMR](https://openreview.net/forum?id=eBMOr6a84z&noteId=CdDdh4nPiQ), [jPXU](https://openreview.net/forum?id=eBMOr6a84z&noteId=JiKjJmW5YE)) Clarified the teacher design (frozen, periodically refreshed copy of the policy, used only as a potential) and added a “Teacher selection” ablation (App. C) showing that this frozen-policy teacher outperforms stronger cross-family teachers (Qwen3-4B-TIPS, Llama3.1-8B), indicating that distributional alignment matters more than raw teacher strength.
* ([vPWX](https://openreview.net/forum?id=eBMOr6a84z&noteId=sw5H21H2ll), [VPap](https://openreview.net/forum?id=eBMOr6a84z&noteId=I4l3TcOFDQ), [ajMR](https://openreview.net/forum?id=eBMOr6a84z&noteId=CdDdh4nPiQ), [jPXU](https://openreview.net/forum?id=eBMOr6a84z&noteId=JiKjJmW5YE)) Added a detailed compute/efficiency analysis (Sec. 4.2, App. H): TIPS adds ≈11–12% training FLOPs and ≈16–17% per-step wall-time over PPO, while GRPO is ≈3.5× PPO in FLOPs; TIPS adds no inference-time cost.
* ([jPXU](https://openreview.net/forum?id=eBMOr6a84z&noteId=JiKjJmW5YE), [VPap](https://openreview.net/forum?id=eBMOr6a84z&noteId=I4l3TcOFDQ), [ajMR](https://openreview.net/forum?id=eBMOr6a84z&noteId=CdDdh4nPiQ), [vPWX](https://openreview.net/forum?id=eBMOr6a84z&noteId=sw5H21H2ll)) Strengthened analysis of training hyperparameters: (i) replaced the coarse ($\alpha$) ablation with a systematic study and dynamic-$\alpha$ experiments (Sec. 4.3, App. D), showing a broad “medium” regime where TIPS is robust; (ii) added a “Refresh interval” ablation for the teacher (Sec. 4.3, App. F), showing that stale teachers hurt performance, while a broad range around 200 steps works well and all moderate intervals still outperform PPO.
* ([EXdd](https://openreview.net/forum?id=eBMOr6a84z&noteId=khNAqnKkLY), [vPWX](https://openreview.net/forum?id=eBMOr6a84z&noteId=sw5H21H2ll), [VPap](https://openreview.net/forum?id=eBMOr6a84z&noteId=I4l3TcOFDQ), [ajMR](https://openreview.net/forum?id=eBMOr6a84z&noteId=CdDdh4nPiQ)) Extended experimental coverage: added results for Qwen2.5-14B-Instruct, Qwen3-4B-Instruct, and Llama3.1-8B (Table 3), showing consistent EM/F1 gains over PPO across model families and scales with similar ≈11–12% FLOPs overhead; added BrowseComp-Plus experiments (App. B) showing TIPS improves PPO/GRPO and that a Qwen3-4B-TIPS agent can outperform a reported 32B search agent under the same BM25 configuration.
* ([EXdd](https://openreview.net/forum?id=eBMOr6a84z&noteId=khNAqnKkLY), [VPap](https://openreview.net/forum?id=eBMOr6a84z&noteId=I4l3TcOFDQ), [ajMR](https://openreview.net/forum?id=eBMOr6a84z&noteId=CdDdh4nPiQ), [vPWX](https://openreview.net/forum?id=eBMOr6a84z&noteId=sw5H21H2ll)) Clarified the experimental setting and baselines (VERL 0.5 multi-turn Search-QA rather than the original Search-R1 code), explained how we re-evaluate other RL search agents under a shared protocol, and added context on the relationship between TIPS, PRMs, and LLM-as-judge methods. Included a brief qualitative discussion of failure modes and an explicit failure example for the information reward (App. G.2, Fig. 12).

We also made smaller improvements to notation, organization, and writing clarity, which we discuss in our responses to each reviewer.

Again, we thank the reviewers for their constructive feedback. We believe that all comments have been addressed in this revision and are happy to discuss any remaining concerns.

Best,

Authors of *TIPS*

---

### Comment · Area_Chair_NhC2 · 2025-11-24
**Reviewer & Author Discussion**

Hi Reviewers,

Please kindly and actively participate in the review-author discussion if you haven't already, raise your further concerns so that the authors can explain more, and make your final decisions.

Best,
AC

---

### Meta-Review · Area_Chair_2nwQ · 2026-01-02

**Summary:**

This paper introduced TIPS, a novel RL framework to overcome the challenges of sparse rewards and difficult credit assignment in training multi-turn, search-augmented LLMs. To address the instability, TIPS implements a dense, turn-level reward mechanism that measures information gain by how much each step increases a teacher model's likelihood of generating the correct answer. By framing this signal as PBRS, the framework provides fine-grained, intermediate feedback. Evaluations across some QA benchmarks demonstrate that TIPS consistently outperforms baselines.

**Reviewer Concerns:**

All the reviewers have a computational complexity concern. Due to the methodology design, it needs additional cost due to the teacher model and the original manuscript did not include this disadvantage. During the rebuttals, the authors provided some quantitative analyses to analyze the additional overhead. Some reviewers are okay with those additional computational complexity (due to the performance gain); however, some other reviewer has claim that marginal performance improvements cannot be compensated by the significant overhead. Some reviewers also raised the points that the experiments should be expanded from Qwen-2.5 from other models as well as additional benchmarks. The authors provided some additional experiments during the rebuttal to reduce the concerns. Lastly, most reviewers raised the concerns on the dependency on the teacher model because teacher model's output is utilized as the part of the rewards. The authors provided additional experiments to reduce the concerns. Overall, AC thinks that the responses resolved most of the concerns on the papers raised by the reviewers.

**Reviewer Scores:**

The initial scores were (4,6,4,4,6). During the rebuttal, two reviewers were engaged. The first engaged reviewer did not change the score from the original score 4 due to the limitations of the TIPS on generalizability and cost overhead. On the other hand, the second engaged reviewer expressed to increase the score from 6 to 8. So, the final score would be (4,6,4,4,8).

---

### Decision · Program_Chairs · 2026-01-26

Accept (Poster)